

# Assessing the benefit of snow data assimilation for runoff modeling in alpine catchments

Nena Griessinger[1,2], Jan Seibert[2], Jan Magnusson[3], and Tobias Jonas[1]

[1]WSL Institute for Snow and Avalanche Research SLF, Davos, Switzerland
[2]Department of Geography, University of Zurich, Zurich, Switzerland
[3]Norwegian Water Resources and Energy Directorate (NVE), Oslo, Norway

*Correspondence to:* Nena Griessinger (nena.griessinger@slf.ch)

**Abstract.** Snow models have been developed with a wide range of complexity depending on the purpose of application. In alpine catchments, snowmelt often is a major contribution to runoff. Therefore, modeling snow processes is important when concerned with flood or drought forecasting, reservoir operation and inland waterway management. In this study, we address the question of whether the performance of a hydrological model can be enhanced by integrating data from a dedicated

external snow monitoring system. As a framework for our tests we used the hydrological model HBV (in the version HBV light), which has been applied in many hydrological studies and is also in use for operational purposes. While HBV originally follows a temperature index approach with time-invariant calibrated degree-day factors to represent snowmelt, in this study the HBV model was modified to use snowmelt time series from an external and spatially distributed snow model as model input. The external snow model integrates three-dimensional sequential assimilation of snow monitoring data with a snowmelt

model, which is also based on the temperature-index approach but uses a time-variant degree-day factor. The following three variations of this external snow model were applied: a) the full model with assimilation of observational snow data from a dense monitoring network, b) the same snow model but with data assimilation switched off, c) a downgraded version of the same snow model representing snowmelt with a time-invariant degree-day factor. Model runs were conducted for 20 catchments at different elevations within Switzerland for 15 years. Our results show that at low and mid elevations the performance of

the runoff simulations did not vary considerably with the snow model version chosen. At higher elevations, however, best performance in terms of simulated runoff was obtained when using the snowmelt time series from the snow model, which utilized data assimilation. This was especially true for snow-rich years. These findings suggest that with increasing elevation and correspondingly increased contribution of snowmelt to runoff, the accurate estimation of snowmelt rates gains importance.

## 1  Introduction

Snowmelt provides a dominant contribution to runoff and groundwater storages in mountainous regions. In such areas, modeling snow processes is crucial for resource management as well as for flood and drought forecasting. Snow accumulates and acts as temporary storage of water that is released as soon as snowmelt occurs. Since erroneous simulations of snow accumulation can bias the amount and timing of simulated snowmelt, accurately modeling both processes is important for runoff predictions. Problems for modelers may occur due to the great heterogeneity and variability of these processes, but also due to the limited





availability of necessary observational data (Adam et al., 2009; Viviroli and Weingartner, 2004; Viviroli et al., 2011). Additionally, computational resources often constrain operational applications as timely model outputs are required. To cope with these challenges, many hydrological models make use of the temperature-index (TI) melt method instead of the energy balance approach, which has higher input data requirements and also is computationally more demanding (Vehviläinen, 1992; Kumar et al., 2013). TI-models can result in sufficient model performance if evaluated at a daily resolution and at the catchment scale (Lang and Braun, 1990; Hock, 2003), provided they use a reasonable parameterization (such as degree-day factor (DDF) and threshold temperature). The basic concept of TI-models is to use air temperature as a proxy for the three energy sources that contribute to snowmelt: incoming longwave radiation, absorbed global radiation, and sensible heat flux (Ohmura, 2001). The methods differ in their number of parameters such as threshold values to parameterize snowfall and melt, ranging from implementations using 2 to 5, as in HBV (Bergström, 1976), to 11 (Irannezhad et al., 2015) parameters. Inappropriate calibration of parameters will fail to accurately describe accumulation and melt rates and lead to a biased duration of the snow season and incorrect melt-out dates (Seibert, 2003). Identifying catchment characteristics that impact hydrological responses (i.e. geology, soil types, or land use types) is also critical (Fontaine et al., 2002). Snow models of high complexity have been developed for a great variety of applications and their development is still ongoing. For avalanche research or snow studies on a small scale, simulating detailed processes within the snowpack is of great interest and importance. Otherwise, for operational purposes, which require short computation time and therefore cannot represent snowpack processes at great detail, different approaches are used to simulate snow accumulation and melt. Recently, various methods to assimilate observational snow data for snow cover models have been developed. At the point scale, model improvements due to assimilation of snow water equivalent data from observations were already shown (Magnusson et al., 2014). At the catchment scale and for operational purposes, only a few studies evaluated the effect of snow data assimilation with different approaches. Franz et al. (2014) evaluated data assimilation based on a small number of ground-based observation sites within a hindcasting framework. In contrast to predictions of runoff under low flow conditions, the overall skill of the forecasts could not be significantly improved. Jörg-Hess et al. (2015) improved snow water and runoff volume predictions by replacing simulated snow water equivalent at model initialization with data from measurements. Integrating snow data sets within the calibration procedures is an additional method to improve hydrological models as shown by Finger et al. (2015). Andreadis and Lettenmaier (2006) showed that the assimilation of remotely sensed snow cover area data did not significantly improve the model performance during accumulation, while for the snowmelt season small improvements were found. The authors concluded that assimilating snow water equivalent data from observations might be a more successful approach. Therefore, in this study, we evaluated the sensitivity of a conceptual runoff model (conceptual in terms of the linear reservoir concept) to the external input of snowmelt data from three different snow models of different complexities. Particularly, we examined the benefit of snow water equivalent data assimilation for hydrological applications in mountainous regions.



## 2 Data

To cover a wide range of elevations and different climatic regions, for this study we chose 20 catchments spread over Switzerland. All of them were at most minimally affected by human activities, such as water regulation or abstraction. A further crucial selection criterion was the availability of the required data. Since, especially at high elevations, the runoff regime of

many catchments in Switzerland is affected by man-made installations, the number of possible catchments was highly limited. Catchments analyzed in this study varied in size from 17 to 473 $km^2$ and the mean elevations of these catchments ranged between 560 and 2656 m.a.s.l. (Table 1 and Figure 1). We grouped the catchments for our analysis based on their mean elevation into three elevation classes: below 1000 m.a.s.l., 1000 to 2000 m.a.s.l., and above 2000 m.a.s.l.. Runoff data measured at the catchment outlets of these 20 catchments was provided by FOEN (Federal Office of the Environment). We used daily

average values for the entire study. For the data assimilation for the full snow model used in this study we considered daily snow depth measurements from both manual and automatic monitoring stations (see red stars in Figure 1 for locations). All 320 stations used were part of either the MeteoSwiss (Federal Office of Meteorology and Climatology) or the SLF (WSL Institute for Snow and Avalanche Research) snow station networks in Switzerland, covering elevations between 210 and 2950 m.a.s.l. and located on open, flat terrain. Daily data from the morning measurements between 01 September 1998 and 31 August 2013

was carefully checked for missing values or erroneous readings and corrected where necessary. Temperature data was obtained from 220 stations and interpolated using an inverse distance weighting approach as described in Magnusson et al. (2014), which considers both, horizontal and vertical distances between measurement stations and interpolated grid cells. A variable weighting factor was used to determine the influence of horizontally near but vertically distant stations. The resolution of the resulting temperature grid dataset was 1 km by 1 km. Precipitation data was also required as a gridded input dataset. We used a

daily product (RhiresD) with a spatial resolution of 2 km by 2 km available from MeteoSwiss. The product is based on a dense precipitation gauge network with approximately 500 stations within Switzerland. Methodological details are described in Frei and Schär (1998), Frei et al. (2006) and Isotta et al. (2014).

## 3 Methods

### 3.1 Hydrological Model

The semi-distributed hydrological model HBV (Bergström, 1976, 1992; Bergström et al., 1995; Lindström et al., 1997) in the version HBV light (Seibert and Vis, 2012) was used to simulate runoff at the 20 selected catchments. HBV requires a time series of precipitation, air temperature, and potential evaporation to simulate runoff for a specific catchment. Potential evaporation was calculated following the methods of Priestley and Taylor (1972). In the HBV snow routine, precipitation is expected to be solid below a certain temperature threshold and multiplied by a correction factor to account for possible undercatch and

to compensate for the missing snow interception. Snowmelt is usually calculated using the same threshold temperature and a DDF. Up to a certain fraction, liquid water can be stored in the snowpack and refreezes if temperatures are below the threshold temperature. In our study, however, we disabled this snow routine of the HBV model and replaced snowmelt as well as rain



input with data from the external snow model. Groundwater recharge and actual evaporation were simulated in a soil routine depending on the actual water storage. A response routine consisting of three linear reservoirs and a routing routine using a triangular weighting function follow. Runoff data observed at the outlet of all catchments considered in this study was used for calibration and validation of the model. More details are available in Seibert and Vis (2012). To evaluate the performance of the

hydrological model in response to the input from different variants of the external snowmelt model, we focused our analysis on the main melt period, denoted below as snowmelt season. Although onset and duration of the snowmelt season vary from year to year, we have determined a fixed snowmelt season for each individual catchment (Table 1), based on the average timing of first snowmelt runoff in the spring and the average duration until 75% of melt-out. Two approaches were chosen to split the available runoff data into separate datasets for calibration and validation. The first approach was to use all years for calibration

except one, which was used for validation. This so called leave-one-out procedure was repeated so that each year was used for validation once. The second approach was differential split-sampling (Klemeš, 1986), where the snow-poor and normal years were used for calibration and the snow-rich years were used for validation. This separation into different snow year groups was done individually for each catchment. To optimize the parameter set of the hydrological model for each catchment and each of the input datasets within the calibration period, we ran a genetic calibration algorithm as described in Seibert (2000) with

5000 model runs and 1000 runs for local optimization. As the objective function, we used the Nash-Sufcliffe model efficiency (Nash and Sutcliffe, 1970) computed for the catchment-specific snowmelt season.

### 3.2 Snow Model

The external snow model framework, which we used in this study instead of the snow routine built in the HBV model, also simulates snowmelt by a TI approach but in addition allows for integration of observational snow data using a data assimilation

scheme. While some details on the external snow model framework are given below, a full description of model and data assimilation methods is available in Magnusson et al. (2014). We applied three versions of this model, denoted M1 to M3. Version M1 includes the full model and data assimilation scheme (an approach unavailable in the internal snow routine of HBV), whereas M2 an M3 are downgraded versions of M1 as described below. Common to all model versions described below is that a threshold temperature differentiates whether precipitation falls as snowfall or rain. However, our models allow

for mixed precipitation in a range close to the threshold temperature (see Equation 10 and the corresponding description in Magnusson et al. (2014)). For all versions, the snow covered fraction (SCF) is simulated using a parameterization developed by Essery et al. (2008), which is derived from the assumption that the snow cover follows a lognormal probability distribution prior to melt (see also Equation 13 in Magnusson et al. (2014)). Further, all the three model versions described below allow the snow cover to hold a fraction of liquid water. In the following section, we describe the three versions of the snow model used

in this study:

- TI snowmelt model with data assimilation and time-varying DDF (M1): This model is the same as described in detail in Magnusson et al. (2014). Using an elaborated TI approach, daily snowmelt at each grid cell was calculated if a certain threshold temperature is exceeded. The DDF defines the possible melt rate per day and per degree temperature above the threshold. For M1, the DDF varied as a function of season between a minimal and maximal value using a sinusoidal function (see Equation 12





in Magnusson et al. (2014)). For the data assimilation, the daily measured snow depth data at all stations were first converted to snow water equivalents (SWE) using a snow density model, which is based on the methods of Jonas et al. (2009) and Martinec and Rango (1991). Second, by applying an optimal interpolation approach, the SWE data was used to correct the computed snowfall amounts. Finally, the simulated melt rates and model state variables (SWE and liquid water content) were

updated using the ensemble Kalman filter with the same SWE data. Both the optimal interpolation scheme and the ensemble Kalman filter were setup using spatially correlated error statistics. With such an approach, often called three-dimensional data assimilation, the point snow observations influences the gridded simulation results even at locations lacking observations. For more details about the model, and the data assimilation method in particular, see Magnusson et al. (2014).

- TI snowmelt model with time-varying DDF without data assimilation (M2): In this version, the same elaborated TI ap-
proach as in M1 was applied to simulate snow accumulation and melt at each grid cell. The DDF seasonal variations are equal to those in M1. However, the data assimilation procedures were switched off such that observed SWE data were not used to update the initial estimates on snow accumulation and melt rates.

- TI snowmelt model using a constant DDF without data assimilation (M3): This version differs from M2 with respect to the DDF. Here the DDF does not show seasonal variations but is assumed to be constant over season. The average DDF of 2.5 mm
°C-1 day-1 was chosen, which is a good compromise if used for the full winter season. For comparison only, complementary analyses were performed with the constant DDF of 4.0 mm °C-1 day-1, which is more appropriate if used for a late snowmelt season only. Note that M3 represent the type of snow routine used in HBV light, except for that DDF is a model parameter determined by calibration in HBV, whereas it is a pre-defined value in M3.

Replacing a TI model with another TI model, and not with an energy-balance or snowpack-physics model, may appear
unusual at first glance. However, if concerned with conceptual hydrological modeling at a daily time scale, the TI model framework used here constituted an ideal testing environment. To provide daily snowmelt rates, the dynamic data assimilation framework within M1 represents current state-of-the art methodology in operational snow hydrological monitoring. Since it accounts for measured snow depletion rates at hundreds of monitoring sites, it provides the best possible input to the hydro-logical model. Even with data assimilation switched off (M2), if validated against snow lysimeter data at daily time steps, the
performance is almost on par with the output of top-notch energy balance models (Magnusson et al., 2015). Only the con-cept of using a constant DDF (M3) could result in a severely downgraded performance, as already seen by Lang and Braun (1990). Hence, the triplet [M1, M2, M3] provides a ranked set of input options, which allows an evaluation of the sensitivity of conceptual hydrological modeling on the input from different types of snow models. This ultimately was the purpose of the study, rather than testing the performance of a specific runoff model (i.e., HBV). As mentioned above, HBV originally uses
a TI snowmelt routine, which is similar to our external model version M3. However, as part of HBV light, the constant DDF is a free parameter to be optimized during calibration. Hence, to provide a benchmark for our performance tests, we also ran the HBV model with the original snow routine switched on. We used these runs as an upper benchmark, since the HBV snow routine was tuned by calibration to allow the maximum possible performance of the runoff model for each individual catch-ment. In contrast, we created a lower benchmark by assuming all precipitation to be rain, i.e., a no-snow-model scenario. These
two benchmarks allowed scaling of the performances, which were achieved when using M1 to M3 to provide input to HBV.





All model variants were run for the whole study period on a daily time step at 1 km spatial resolution. During the snowmelt season, the three snow model methods created individual spatial pattern of simulated snowmelt. As an illustrative example, the cumulative sums of snowmelt between 01 February 2007 and 30 April 2007 are shown in Figure 2. As expected for the snowmelt season, M2 yielded higher amounts of snowmelt compared to M3 due to differences in the DDF. In this particular

year, the observations used for the assimilation did not support the high melt rates as predicted by M2, resulting in M1 to calculate lesser amounts of snowmelt.

### 3.3 Validation methods

Timing of snowmelt onset and of runoff events due to snowmelt affects the availability of water resources and influences flooding and droughts (Semmens and Ramage, 2013). Therefore, it is crucial to simulate and to evaluate the timing of streamflow

accurately when comparing snowmelt models. Several efficiency criteria are used in the literature for evaluating hydrological models and should be selected carefully depending on the aim of the validation (Krause et al., 2005). To assess the performance of the hydrological model in combination with the input options from our set of snow models, we chose the following two criteria. First, since we were interested in how precise single peak flow events due to snowmelt could be simulated when integrating data from the different snow model approaches, we used the "Peak flow Efficiency for snowmelt season" $E_{PF}$. Fig-

ure 3 illustrates the procedure to calculate this measure. Observed peak flow events during the snowmelt season (yellow period in Figure 3) that exceed a certain threshold (defined as 1.5 times of the mean runoff during snowmelt season; horizontal line in Figure 3) were picked and denoted as $Q_{peak\,obs\,i}$ (blue circles in Figure 3). The maximum simulated runoff in a time window of one day before and after each of the $n$ observed peak flow events were taken as simulated reference values $Q_{peak\,sim\,i}$ (red stars in Figure 3). These values did not necessarily have to be local peaks or greater than a certain threshold (Eq. (1); Seibert

20   (2003)).

$$E_{PF} = 1 - \frac{\sum_{i=1}^{n} |Q_{peak\,obs\,i} - Q_{peak\,sim\,i}|}{\sum_{i=1}^{n} Q_{peak\,obs\,i}}, \tag{1}$$

Additionally, the frequently used Nash-Sutcliffe efficiency of runoff $E_Q$ (Eq. 2) according to Nash and Sutcliffe (1970), which is also supposed to be sensitive to peak flow events (Krause et al., 2005) was chosen and applied to the defined snowmelt season.

$$E_Q = 1 - \frac{\sum_{i=1}^{m} (Q_{obs\,i} - Q_{sim\,i})^2}{\sum_{i=1}^{m} (Q_{obs\,i} - mean(Q_{obs}))^2}, \tag{2}$$


where $i$ represents all (1 to $m$) days within the snowmelt season and $Q_{obs\,i}$ and $Q_{sim\,i}$ are observed and simulated runoff at day $i$, respectively. This was also used as the objective function for the genetic calibration algorithm (GAP-optimization) within the hydrological model framework.





## 4   Results and Discussion

Both efficiency metrics were calculated for *a)* each catchment and *b)* each of the two calibration experiments. The performance statistics are discussed separately for each of the three groups of catchments depending on mean elevation.

### 4.1   Example of runoff simulation for a representative catchment

To look for differences between the three snow model methods, individual catchments and years were selected. Representing a catchment at high elevations, results for the Dischma catchment (EZG 2327, gauge Davos Kriegsmatte) with a mean elevation of 2349 m.a.s.l. are shown in Figure 4. The yellow background displays the catchment-specific snowmelt season during which the bulk of the snowmelt typically occurs. The blue and grey lines at top of the graph indicate the snowmelt input to the hydrological model from M1 excluding and including rain, respectively, in this example for the record-high snow year 1999.
The observed runoff is shown by the black curve, while the different colored curves indicate the simulations with M1, M2 and M3. The curves as well as the performance metrics achieved by the differential split-sample experiment demonstrate that for this catchment, the M1 model as input to the hydrological framework provided the best runoff simulations. Note however, that in this example M1 particularly outperforms the other models in the month of July, which is outside the standard evaluation period.

### 4.2   Model performance across elevation classes: leave-one-out sample

First, we used the leave-one-out approach to calibrate the hydrological model. The leave-one-out approach represents a typical scenario in operational conceptual runoff modeling, i.e. to use as much data as possible for calibration and to apply the resulting parameter values to the current season. Results grouped according to mean catchment height are presented below (Figure 5). Using this calibration procedure, for catchments with mean elevation below 1000 m.a.s.l. the hydrological model showed good
results independent of which snow model was used as input to the hydrological model framework. Even without using a snow model at all (i.e., the lower benchmark), the runoff model resulted in lower but still positive performance values, indicating that the choice of snow model within a conceptual runoff modeling framework is of less importance when dealing with catchments at lower elevations. Similarly for catchments with mean elevation between 1000 and 2000 m.a.s.l. the differences between the three model runs were small. While $E_{PF}$ levels were maintained relative to our assessment for catchments below 1000 m.a.s.l.,
they were separated more clearly from the benchmark model runs, which dropped in performance. $E_Q$ values, on the other hand, decreased for all the M1, M2, M3 and the benchmark model runs. Only for the highest elevation class did the results based on M1 outperform the other model runs, and even reached better $E_{PF}$ values than most simulations at lower elevation classes. Even the model runs based on M2 performed better than those based on M3. This shows that the benefit of better snowmelt input data for conceptual runoff modeling only seems to pay off if considering catchments above a certain elevation. At lower
elevation, differences between the model input options could be mitigated by way of the calibration procedure. Looking at all elevation classes, the median performance of the M1 runs was always higher than the upper benchmark. This was also mostly the case for M2 and M3. This result shows that all versions of the external snow model performed unexpectedly well



in combination with the hydrological framework even though they were not included in the calibration procedure. However, while results based on M1 showed a relatively constant performance across all elevation classes in both $E_{PF}$ and $E_Q$, this was not the case for results based on M2 and M3, which deteriorated with increasing elevation. The above results demonstrate a benefit of using an advanced snowmelt modeling system in the context of conceptual hydrological modeling, even if the benefit

seems comparably small and restricted to catchments above a certain elevation. Other studies that evaluated the influence of integrating snow water equivalent data into hydrological models showed similar results (Finger et al., 2015; Jörg-Hess et al., 2015). Only a few studies have used direct assimilation of ground based snow data. Due to limited availability of ground observations, assimilating remotely sensed snow data is a more common practice but requires further inversion methods, which is quite challenging to implement and induces additional uncertainties (Andreadis and Lettenmaier, 2006). Several studies used

satellite observations of snow cover extent in different assimilation schemes to update snow models. Clark et al. (2006) as well as Thirel et al. (2013) could slightly improve runoff predictions by assimilation of snow covered area using the Ensemble Kalman filter and the particle assimilation filter, respectively. As in the above studies, we focused on a catchment specific snowmelt season and used two performance measures that evaluated the ability of the models to capture peakflow events, among other characteristics of the hydrograph. Simulating such events is of great importance, especially for operational flood

forecasting purposes. While the performance of well-calibrated models may be adequate independent of model complexity (Hock, 2003; Magnusson et al., 2015), we are particularly interested in the model performance in extreme years, when the snowmelt contribution greatly increases flood risks. This is why in the second set of modeling experiments we singled out snow-rich years as validation dataset to generate both, a more challenging and more relevant test scenario.

### 4.3   Model performance across elevation classes: differential split-sample

For the differential split-sample approach, snow-rich years were used to validate the runoff models. As expected, the analysis using the differential split-sample approach revealed similar performance patterns compared to the leave-one-out approach, but with increased differences between model runs (Figure 6). As seen before, at low and mid elevation classes the differences between the three model versions as well as between calibration and validation were comparably small. The median values of efficiencies for each model version ranged between 0.7 and 0.8 ($E_{PF}$) respectively 0.75 and 0.85 ($E_Q$). As seen before, at

high elevations, model results based on M1 were superior to those based on M2, which in turn outperformed the model runs based on M3. However, the differences between the three runs were considerably larger than those seen with the leave-one-out approach. Another notable difference between both calibration methods was that the differential split-sample approach led to significantly higher $E_Q$ for validation years compared to calibration years, while the opposite was the case when using the leave-one-out approach. Both findings strongly suggest that the benefit of advanced snowmelt input data for conceptual runoff

modeling is particularly valuable in situations that feature a strong snowmelt component (high elevation, snow-rich years). Both $E_{PF}$ and $E_Q$ for M1-based model runs show an exceptional performance at high elevation for validation years, which highlights the value of snow data assimilation when concerned with forecasting snowmelt related floods. An additional analysis was performed with M3 using a DDF of 4.0 mm °C-1 day-1 (results not included in figures). This is a typical value found in the literature for high elevations with melting conditions later in the season (Martinec et al., 1983). As expected, compared to



the standard DDF of 2.5 mm °C-1 day-1 in M3, the additional model runs resulted in slightly better performance metrics at high elevations with later onset of snowmelt (catchments above 2000 m.a.s.l.), but considerably worse performance in all other model runs.

### 4.4 Model performance for high elevation catchments: leave-one-out sample

The validation of the differential split-sample experiment showed that the three external snow models provided the best runoff simulations for snow-rich years, specifically for catchments with a mean elevation of above 2000 m.a.s.l.. In a further analysis, we ordered the single validation years individually by catchment for the leave-one-out approach from snow-poor to snow-rich based on peak SWE. This procedure allowed testing of whether there was a trend in the runoff performance metrics associated with the snow amount of single years. Such a trend was indeed evident, as seen in Figure 7. Independent of the snow model

used, the best results were achieved when validating the model performance during snow-rich years regarding both $E_{PF}$ and $E_Q$. The performance measures discussed above were computed for a catchment-specific pre-defined fix snowmelt season, which was based on the typical timing of observed snowmelt runoff. While this approach allowed us to focus on the sensitivity of runoff modeling to different approaches for estimating snowmelt, it has four main implications to the interpretation of the results. First, $E_Q$ values tend to be lower if calculated over a short period, and values may not be comparable to $E_Q$ data from

assessment of multi-year or multi-season datasets, in particular if analyzing daily runoff data that do not encompass diurnal variations. Second, within a pre-defined season, the variation of a time-varying DDF as used in M2 is small. Especially at low elevations and early in the year, the DDF of M2 and M3 do not differ much and therefore produces similar runoff simulations with comparable performance. According to Lang and Braun (1990) and Magnusson et al. (2015), a clearer benefit of using a flexible instead of a fixed DDF would have been expected if used within a longer time window. Third, at low elevations

snowmelt may occur sporadically and not necessarily within a pre-defined season. At high elevations, it is also possible that the main melt does not occur within the catchment-specific snowmelt season due to longer melt-out duration of extremely snow-rich years. Consequently, if snowmelt occurred outside of the validation period, it would not affect the performance statistics. This may have partly suppressed differences between the three different snow models. Finally, note that seasonal $E_{PF}$ and $E_Q$ statistics are two metrics out of several possible evaluation criteria. While we also tested other metrics, those

were not further integrated to the discussion, given that the results were similar compared to the performance data presented above.

### 5   Conclusions

Based on daily runoff data measured over a period of 15 years at 20 catchments in Switzerland, we evaluated the sensitivity of a conceptual hydrological modeling framework to snowmelt input from snow models of different complexity. The most complex

snow model integrated three-dimensional sequential assimilation of snow monitoring data with a snowmelt model based on the temperature index approach. In contrast, the simplest snow model represented snowmelt with a constant degree-day factor, and did not include any data assimilation. The snow models were combined with the HBV light hydrological model (Seibert



and Vis, 2012) to produce a runoff record. The performance of the HBV runs based on snowmelt data from the snow models was assessed by way of performance metrics evaluated during the snowmelt season only. Our results showed that advanced methods to calculate snowmelt as input to conceptual runoff models only improved model performance if considering snow-dominated catchments. At low elevations, differences between the model input options were found to be minor. For higher

5  elevation catchments, however, snowmelt input from the data assimilation framework consistently provided the best results. Further analysis demonstrated considerably higher performance metrics for snow-rich years as compared to years with little snow. In contrast to earlier studies, which have shown that assimilation of snow covered area only has limited impact on runoff simulations, our results indicate that the assimilation of snow water equivalent data can have a larger benefit for accurate stream flow predictions. This finding highlights the value of choosing appropriate snow data assimilation methods, and perhaps

10  even more important, selecting the correct variable for assimilation when concerned with operational forecasting of snowmelt related floods.

*Acknowledgements.* This study was partly funded by the Federal Office of the Environment (FOEN). We thank MeteoSwiss for access to the meteorological data and FOEN for providing river runoff observations used in this study. Thanks to Manfred Stähli and Massimiliano Zappa for helpful discussions and to Nathalie Chardon for reviewing the English of this article.





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





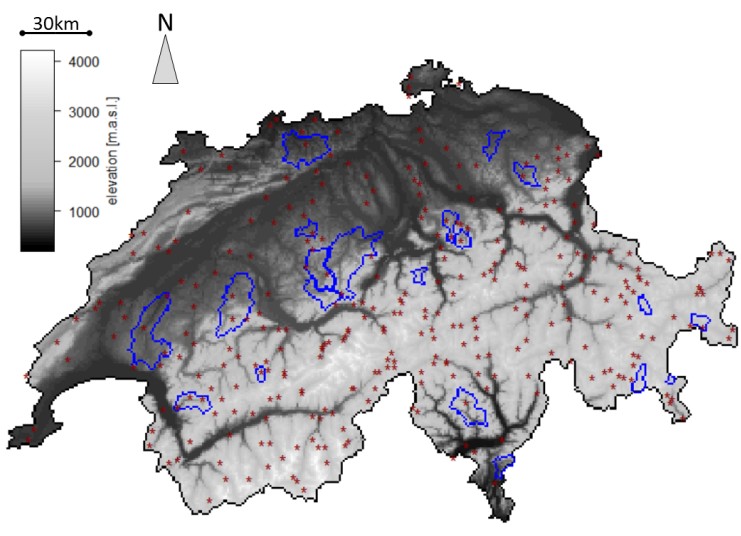

**Figure 1.** Locations of snow observation stations (red stars) and 20 studied catchments (blue border lines) in Switzerland.

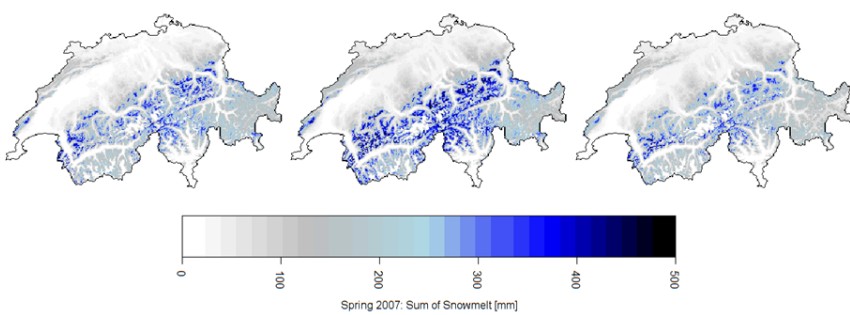

**Figure 2.** Cumulative snowmelt during the snowmelt season 2007 as calculated by the snow model method M1 (full model with data assimilation, left), M2 (full model without data assimilation, middle), and M3 (simplified model, right). The sums between the three model methods differ depending on the use of observational snow data assimilation and the use of different DDFs.





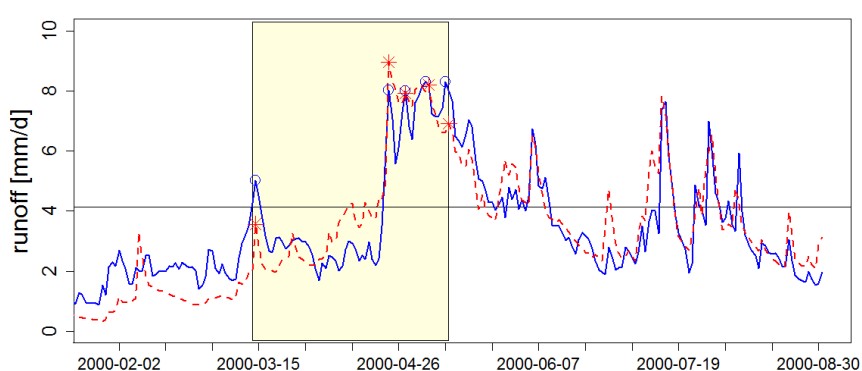

**Figure 3.** Graphical explanation of how to calculate $E_{PF}$. The yellow background shows a catchment-specific snowmelt season window within which the efficiency criteria were computed. The horizontal line indicates the threshold of 1.5 times the mean observed runoff (blue line) above which measured peak flow events (blue circles) are detected. Red stars present corresponding events of the simulated runoff (dashed red line). See Sect. 3.3 for details.





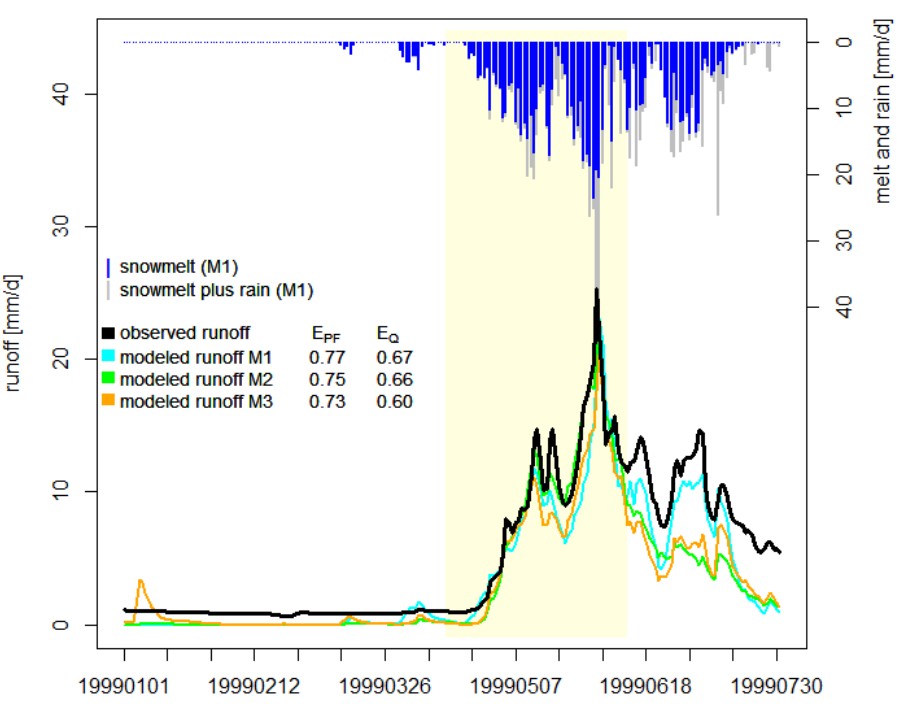

**Figure 4.** Observed and modeled runoff for the Dischma catchment for year 1999, as well as water input from snowmelt and rain modeled with method M1.




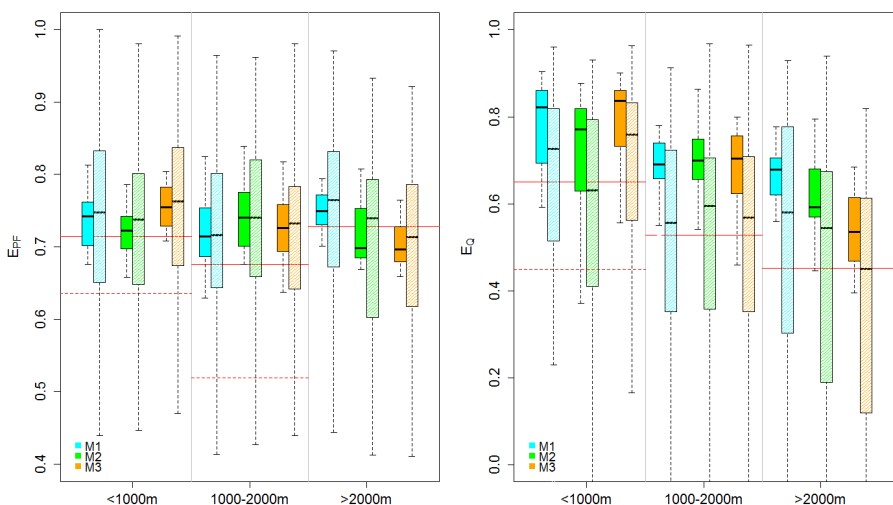

**Figure 5.** Results of the leave-one-out approach. $E_{PF}$ (left panel) and $E_Q$ (right panel) for each elevation class and snowmelt model. For the individual elevation classes and melt models, the left box plots (darker colors) show the results for the calibration period, and the right box plots (lighter colors) show the results for the validation period. The whisker boxes represent the median (center line), the interquartile range (25-75th percentile; box outline) and highest/lowest performance within the interquartile range +/- 1.5 times of the interquartile range (whiskers). The benchmark performance is denoted by a solid red line (upper benchmark) and a dashed red line (lower benchmark), and the latter only displayed if within the range of the axis limits.




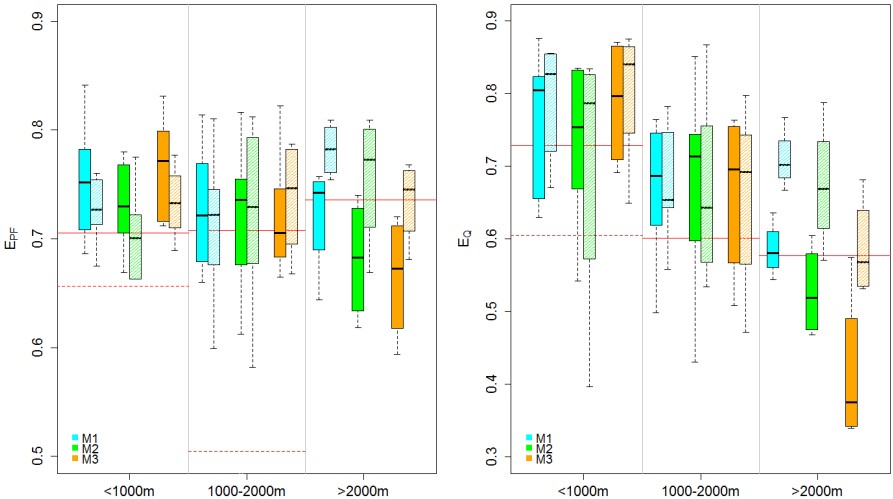

**Figure 6.** Results of the differential split-sample approach. $E_{PF}$ (left panel) and $E_Q$ (right panel) for each elevation class and snowmelt model. For the individual elevation classes and melt models, the left box plots (darker colors) show the results for the calibration period, and the right box plots (lighter colors) show the results for the validation period. The whisker boxes represent the median (center line), the interquartile range (25-75th percentile; box outline) and highest/lowest performance within the interquartile range +/- 1.5 times of the interquartile range (whiskers). The benchmark performance is denoted by a solid red line (upper benchmark) and a dashed red line (lower benchmark), and the latter only displayed if within the range of the axis limits.

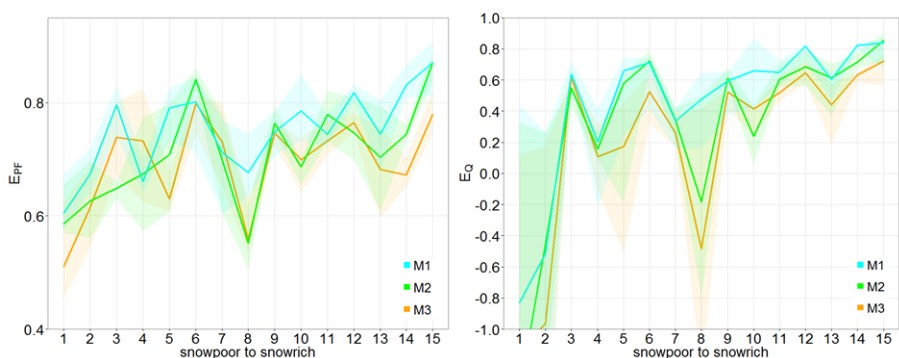

**Figure 7.** Results of the leave-one-out approach for catchments with mean elevation above 2000 m.a.s.l. Median (solid lines) and interquartile (25-75th percentile, shading) range of $E_{PF}$ (left panel) and $E_Q$ (right panel) for validation years ordered from snow-poor (index=1) to snow-rich (index=15) years.





**Table 1.** Characteristics of 20 Swiss catchments in this study.

| Number | Area [$km^2$] | Min elevation [$m.a.s.l.$] | Max elevation [$m.a.s.l.$] | Mean elevation [$m.a.s.l.$] | Elevation class | Begin snowmelt | End snowmelt |
|---|---|---|---|---|---|---|---|
| EZG 2202 | 276 | 305 | 1087 | 577 | 1 | 01-01 | 03-01 |
| EZG 2126 | 77 | 501 | 911 | 640 | 1 | 01-14 | 03-14 |
| EZG 2034 | 416 | 450 | 1402 | 721 | 1 | 01-14 | 03-14 |
| EZG 2343 | 61 | 592 | 1032 | 757 | 1 | 01-14 | 03-14 |
| EZG 2374 | 89 | 649 | 1359 | 948 | 1 | 02-14 | 04-14 |
| EZG 2321 | 74 | 286 | 1809 | 954 | 1 | 02-14 | 04-14 |
| EZG 2603 | 188 | 699 | 1695 | 1040 | 2 | 02-21 | 04-21 |
| EZG 2634 | 473 | 440 | 2261 | 1044 | 2 | 02-21 | 04-21 |
| EZG 2179 | 355 | 609 | 2028 | 1072 | 2 | 03-01 | 05-01 |
| EZG 2609 | 82 | 845 | 1577 | 1096 | 2 | 02-21 | 04-21 |
| EZG 2409 | 127 | 770 | 2007 | 1296 | 2 | 02-21 | 04-21 |
| EZG 2300 | 59 | 918 | 1994 | 1345 | 2 | 03-07 | 05-07 |
| EZG 2203 | 130 | 579 | 2830 | 1546 | 2 | 03-14 | 05-14 |
| EZG 2605 | 188 | 546 | 2590 | 1656 | 2 | 03-14 | 05-14 |
| EZG 2276 | 43 | 931 | 2682 | 1794 | 2 | 03-14 | 05-14 |
| EZG 2232 | 31 | 1360 | 2587 | 1907 | 2 | 03-14 | 05-14 |
| EZG 2366 | 17 | 1920 | 3005 | 2316 | 3 | 04-14 | 06-14 |
| EZG 2304 | 56 | 1797 | 2903 | 2337 | 3 | 04-14 | 06-14 |
| EZG 2327 | 42 | 1772 | 2869 | 2349 | 3 | 04-14 | 06-14 |
| EZG 2256 | 67 | 1833 | 3721 | 2686 | 3 | 05-01 | 07-01 |