# Peer review of "Assessing the benefit of snow data assimilation for runoff modeling in alpine catchments"

_Hydrology and Earth System Sciences, 2016_

## Referee Comment (RC1) · J. Parajka (Referee) · 29 Mar 2016

General comments

The study evaluates the value of external snow distributed input into a conceptual hydrologic model in alpine basins. Three different settings are compared for 20 basins in Switzerland. The results show that assimilation of snow improves the runoff model efficiency in basins with mean basin elevation above 2000 m a.s.l.

This is a nice compact study, I enjoyed reading it. The manuscript is clearly written, has a good structure and it is within the scope of the journal. I have only a few minor comments which might be considered for revision. These include:

1) Introduction: I believe, there are some more relevant studies looking on the benefits

of additional snow data in hydrologic model calibration or modelling. Please consider to extend the introduction section accordingly. (please see e.g. Udnaes et al.,2007, Parajka et al. 2007, 2008, or review in Parajka and Blöschl, 2012)

2) Objectives: Is it the sensitivity (or runoff model efficiency) of conceptual hydrologic model to snow inputs, which is the main objective?

3) P.3, l.9-10: "daily average values for the entire study"? Please clarify.

4) Results: It seems that the way DDF is estimated does affect the performance. Please consider to provide/discuss more detailed information about the sinusoidal function and snow density model used. Does it change with the elevation of the basins?

5) Model efficiency: I would suggest to consider extending results and showing also runoff model efficiency (NSE) for the entire calibration/validation periods (not only the selected snowmelt seasons). This might serve as a baseline for comparison with other studies, as well as to allow to evaluate the value of improved snowmelt for the following seasons (e.g. are the soil moisture states/and hence runoff generation different for the three variants?).

6) Figure 2: Please consider to decrease the legend and increase the size of the maps.

7) Table 1. Please give names to basins.

References:

Udnaes, H. Ch., Alfnes, E., and Andreassen, L. M. (2007). Improving runoff modeling using satellite-derived snow cover area? Nordic Hydrology, 38(1), 21-32

Parajka , J.,Merz, R. and Blöschl, G. (2007) Uncertainty and multiple objective calibration in regional water balance modelling: case study in 320 Austrian catchments. Hydrological Processes, 21 (4), 435-446.

Parajka, J. and Blöschl, G. (2008). The value of MODIS snow cover data in validating and calibrating conceptual hydrologic models. Journal of Hydrology, 358, 240–258.

Parajka, J. and G. Blöschl (2012) MODIS-based Snow Cover Products, Validation, and Hydrologic Applications, In: Eds (Ni-Bin Chang) Multiscale Hydrologic Remote Sensing: Perspectives and Applications, Chapter 9, CRC Press, 550 pp..

---

## Referee Comment (RC2) · Anonymous Referee #2 · 15 Apr 2016

General Comments

The work of Griessinger et al. assesses the added value (or lack thereof) of a hierarchy of complexities in degree-day snow-models, possibly including SWE data assimilation. This type of models is frequently used in hydrological modelling.

This manuscript is of high value for hydrological modellers in snow-dominated and snow-influenced catchments, and draws important conclusion as to the desirable level of complexity to be chosen depending on the type of catchment concerned (high snow/enduring-snow cover, low snow depth /ephemeral snow). The different model versions used here build a clever set-up to test the impact of different snow-melt parametrizations and of SWE data assimilation within a hydrological model.

However, a few important considerations are missing, which would strengthen the conclusions of the paper. These are listed below :

- First, the uncertainty associated with snow depth observation data is never mentioned. As I understand from the manuscript the collected snow depth data were rather punctual and to me, the mentioned 'flatness' of the terrain where they were collected does not guaranty their 'local' representativity. Elaboration on that, and precisions as to the snow depth measurement protocol, would be welcome. An ancillary aspect also regards the hydrological data, which are subject to quite high uncertainties in mountain catchments as a result of frequent shifts in the topography of the river beds. This aspect should at least be discussed.

- Second, in most calibration and validation sets of simulations, M3 outperforms the upper-benchmark, which relies on a calibrated degree-day factorn whereas M3 relies on a constant degree-day factor for all catchments. To me this result is quite counter-intuitive and deserves an explanation.

- Finally, a distinct 'discussion' part could be inserted in the manuscript : Section 4.4 after line 11 could be part of it, as well as elements coming in response to point 2 mentioned above. Optionally, more elements as to the different, converging metrics used could be provided to the reader. The general decrease of (each) model performances with elevation could be commented and interpreted, in link with the quality of the interpolations (/extrapolation) of meteorological data and sometimes snow observations at these altitudes.

Minor Comments

- The last sentence of the abstract overlooks the fact that with altitude, not only the accurate estimation of snowmelt rate gains importance, but also the accurate estimation of SWE, which is one of the hypotheses tested by the paper's set-up.

---

## Referee Comment (RC3) · O. Rössler (Referee) · 28 Apr 2016

The article addresses a highly relevant topic that is the added value of implementing external snow models and snow data in a hydrological model. In the present study, three different snow models of increasing complexity are attached to a HBV model and tested in 20 different mesoscale catchment within Switzerland. The catchment cover all altitudes present in the Alps. For all catchments the model performance of reproducing the runoff with in the snow melt seasons was assessed and served as the basis to judge the added value of the snow models. The authors found that the implementation of a snow model that additionally assimilates observed SWE data improves the runoff considerably, especially in high altitudes and in snow-rich years. The article is very well structured and written, concise and comprehensive at the time. The article is to my knowledge of original content and suits well in the scope of the journal. I still need

to point out some more general concerns and a couple of minor comments. After a revision of the manuscript that take into account this concerns and comments, I would recommend publication:

superior comments/concern: a) The first concern addresses the interpretation of the results. What is exactly the added value of the assimilated data set. Is it a more sophisticated and correct snow melt model or is it rather the added indirect information of precipitation amounts fallen in high altitudes where the meteorological station network is not present. My interpretation would be the latter, as the differences between model M1 and M2 (e.g. assimilation) are considerable for the highest altitudes. I would appreciate a discussion on this question. b) A follow up on this issue. The SLF station data are known to overestimate the SWE amounts. How was this issue addressed in the study and if not what are the consequences for your model as you may have calibrated your model against "differently wrong" data. c) The LOO validation produces by nature highly variable performance values. I find it difficult to estimate differences between the models based on medians of boxplot. I would rather use a significance test. I recommend to show validation boxplots side by side and add notches to them. d) I found examples on the model performance given in Figure 3 and 4 show some room for improvements. Especially in Figure 3 it seems as the threshold for snowmelt was calibrated incorrectly. Is this threshold predefined by the external snowmodel? And if so, doesn't this mean that the snow model itself needs to be updated and calibrated against discharge? And I wonder what the upper benchmark model would look like.

special comments/questions: Page 1 Line 1: Abstract: The first sentence is somehow isolated from the rest of the text. I recommend to delete this sentence P2 L1: and the erroneous precipitation input data at higher altitudes? P3 L 32 "rain input" : which precipitation data set drives the snow model? Also the RHiresD? P4 L1 ff: Is it correct that all model combinations HBV+M1-M3 as well as upper and lower benchmark models are calibrated? This is somehow suggested by Figure 5. In the calibration section I understood that a calibration was done for M3, upper and lower benchmark.

P5 L 2-3 what do you mean by "optimal interpolation approach". What magnitude of summed corrections can be found? P5 L12: ….. , but the RHiresD precipitation data set. Correct? P7 L12ff and Figure4: However, the differences between M1-M3 are rather small for the snowmelt season as also indicated by the differences in NSE P7 L27: I wonder if the differences of the LOO validation are significant given the relatively large spread. (see general comments) P7 L31 and Figure 5: - The benchmark lines are only the median of their respective boxplots? What is the spread of benchmark models? - The only difference between the benchmark model and M3 is a predefined DDF in M3 (cp.P5, L17-18)? Or are there further differences? If not, it is unexpected to see M3 to reach higher performance values then the upper benchmark. - Why is the performance of the benchmark model so weak in comparison to the other models especially in the lowest catchment class where snow does not really play a role? P8 L17: Please specify snow-rich: extreme snow years do not necessary result in an increased flood risks. To my understanding, largest snow melt contribution to runoff is expected if snow-covered area is largest and snow depth is widely insignificant (if SWE is above a certain minimum). P8 L30: in snow rich years the extent of snow in the lowlands is presumable larger then in snow-poor years. Accordingly, I also expected an effect of snow-rich years in the lowlands? Can you comment on this? P9 Conclusion: see superior comments Figure 1: The blue lines on black are nearly invisible. Please change colors. Figure 2: Instead of showing one specific year, I would rather see a mean snow melt sum. In addition, maps showing differences between the models would increase readability. Figure 3: Please indicate which model version is represented by the red dashed line. Figure 4: Please add upper benchmark model Table 1: Instead of numbers I would prefer to see the names of the catchments

---

## Author Comment (AC1) · 29 Apr 2016

Dear Juraj Parajka,

we would like to thank you for the positive feedback and the useful comments.

Please find below our replies as inserted blue text.

Kind regards,

Nena Griessinger, Jan Seibert, Jan Magnusson and Tobias Jonas

General comments

The study evaluates the value of external snow distributed input into a conceptual hydrologic model in alpine basins. Three different settings are compared for 20 basins in Switzerland. The results show that assimilation of snow improves the runoff model efficiency in basins with mean basin elevation above 2000 m a.s.l.

This is a nice compact study, I enjoyed reading it. The manuscript is clearly written, has a good structure and it is within the scope of the journal.

I have only a few minor comments which might be considered for revision. These include:

1) Introduction: I believe, there are some more relevant studies looking on the benefits of additional snow data in hydrologic model calibration or modelling. Please consider to extend the introduction section accordingly. (please see e.g. Udnaes et al.,2007, Parajka et al. 2007, 2008, or review in Parajka and Blöschl, 2012)

Thank you for the suggested literature which we will include.

2) Objectives: Is it the sensitivity (or runoff model efficiency) of conceptual hydrologic model to snow inputs, which is the main objective?

We will adapt the manuscript to clarify our objective, how different methods for simulating the snow cover influence runoff predictions.

3) P.3, l.9-10: "daily average values for the entire study"? Please clarify.

Thank you, we will clarify the description.

4) Results: It seems that the way DDF is estimated does affect the performance. Please consider to provide/discuss more detailed information about the sinusoidal function and snow density model used. Does it change with the elevation of the basins?

Thank you; DDF does not change with the elevation of the basins. To clarify the text which obviously raised questions, we will edit this section.

5) Model efficiency: I would suggest to consider extending results and showing also runoff model efficiency (NSE) for the entire calibration/validation periods (not only the selected snowmelt seasons). This might serve as a baseline for comparison with other studies, as well as to allow to evaluate the value of improved snowmelt for

the following seasons (e.g. are the soil moisture states/and hence runoff generation different for the three variants?).

The main objective of this study is to assess how different methods for simulating the snow cover influences runoff predictions. Therefore, we focus on a period spanning 60 days that is strongly affected by snowmelt. We agree that improved modeling of snowmelt might also affect runoff model performances later in the year. An analysis of the leave-one-out experiment for a snowmelt period of 120 days (see below Figure 1 and Figure 2) gave similar results as those shown in our study. The performances are generally lower than within the studied 60 days of snowmelt. We will point out and discuss this finding in the revised version.

6) Figure 2: Please consider to decrease the legend and increase the size of the maps.

Thank you, we will adapt the mentioned figure.

7) Table 1. Please give names to basins.

Thank you, we will adapt the mentioned table.

References:

Udnaes, H. Ch., Alfnes, E., and Andreassen, L. M. (2007). Improving runoff modeling using satellite-derived snow cover area? Nordic Hydrology, 38(1), 21-32

Parajka , J.,Merz, R. and Blöschl, G. (2007) Uncertainty and multiple objective calibration in regional water balance modelling: case study in 320 Austrian catchments. Hydrological Processes, 21 (4), 435-446.

Parajka, J. and Blöschl, G. (2008). The value of MODIS snow cover data in validating and calibrating conceptual hydrologic models. Journal of Hydrology, 358, 240–258.

Parajka, J. and G. Blöschl (2012) MODIS-based Snow Cover Products, Validation, and Hydrologic Applications, In: Eds (Ni-Bin Chang) Multiscale Hydrologic Remote Sensing: Perspectives and Applications, Chapter 9, CRC Press, 550 pp..

[Figure]

**Figure 1: Results of the leave-one-out approach calculated for 120 days of melt.** $E_{PF}$ (left panel) and $E_Q$ (right panel) for each elevation class and snowmelt model. For the individual elevation classes and melt models, the left box plots (darker colors) show the results for the calibration period, and the right box plots (lighter colors) show the results for the validation period. The whisker boxes represent the median (center line), the interquartile range (25-75th percentile; box outline) and highest/lowest performance within the interquartile range +/- 1.5 times of the interquartile range (whiskers). The benchmark performance is denoted by a solid red line (upper benchmark) and a dashed red line (lower benchmark), and the latter only displayed if within the range of the axis limits.

[Figure]

**Figure 2: Results of the leave-one-out approach calculated for 120 days of melt for catchments with mean elevation above 2000 m.a.s.l.** Median (solid lines) and interquartile (25-75th percentile, shading) range of $E_{PF}$ (left panel) and $E_Q$ (right panel) for validation years ordered from snow-poor (index=1) to snow-rich (index=15) years.

---

## Author Comment (AC2) · 29 Apr 2016

Dear Reviewer,

we are very thankful for the positive feedback and the useful comments.

Please find below our replies as inserted blue text.

Kind regards,

Nena Griessinger, Jan Seibert, Jan Magnusson and Tobias Jonas

General Comments

The work of Griessinger et al. assesses the added value (or lack thereof) of a hierarchy of complexities in degree-day snow-models, possibly including SWE data assimilation. This type of models is frequently used in hydrological modelling. This manuscript is of high value for hydrological modellers in snow-dominated and snow-influenced catchments, and draws important conclusion as to the desirable level of complexity to be chosen depending on the type of catchment concerned (high snow/enduring-snow cover, low snow depth /ephemeral snow). The different model versions used here build a clever set-up to test the impact of different snow-melt parametrizations and of SWE data assimilation within a hydrological model.

However, a few important considerations are missing, which would strengthen the conclusions of the paper. These are listed below:

- First, the uncertainty associated with snow depth observation data is never mentioned. As I understand from the manuscript the collected snow depth data were rather punctual and to me, the mentioned 'flatness' of the terrain where they were collected does not guaranty their 'local' representativity. Elaboration on that, and precisions as to the snow depth measurement protocol, would be welcome. An ancillary aspect also regards the hydrological data, which are subject to quite high uncertainties in mountain catchments as a result of frequent shifts in the topography of the river beds. This aspect should at least be discussed.

We will include a discussion about the representativeness and uncertainty of the used punctual snow depth data. The stations used in this study were chosen carefully avoiding sites that are clearly influenced by strong wind drift of snow or frequent sensor failures. Similar datasets were already used for previous studies (Joerg-Hess et al. 2014, Magnusson et al, 2014). Since the dataset is used for operational monitoring of snow water resources, the data has been checked for erroneous or missing data. Measurement records taken at the same station at previous or following days and those taken at neighboring stations were used to appropriately replace or fill the measurement gaps. Regarding the runoff measurements, we rely on the quality of the data provided by the Federal Office of the Environment (FOEN). Nevertheless, we checked the data for missing values.

- Second, in most calibration and validation sets of simulations, M3 outperforms the upper-benchmark, which relies on a calibrated degree-day factorn whereas M3 relies on a constant degree-day factor for all catchments. To me this result is quite counter-intuitive and deserves an explanation.

Thank you for this remark, we agree that the finding M3 to outperform the upper benchmark may appear counterintuitive and should be discussed.

Dealing with liquid water content, refreezing, cold content dynamics and the partitioning of rain and snow are - among others - elements that influence the performance of temperature-index models. These elements differ between [M1, M2, M3] and HBV. In [M1, M2, M3] the representation of those processes have been particularly trained for optimal performance in the Swiss Alps.

Further, calibrating HBV for the melt season only – as done in our study – could result in a DDF that is too high during the snow accumulation period. The consequence might be an unbalanced performance with good snowmelt rates during the melt season at the price of too little accumulation earlier in the year with unwanted side effects on the snow depletion dynamics. M3 features a more moderate DDF of 2.5 $mm°C^{-1}day^{-1}$ allowing for a more balanced performance over the entire snow season.

We will adapt the manuscript and provide this discussion to the reader.

- Finally, a distinct 'discussion' part could be inserted in the manuscript : Section 4.4 after line 11 could be part of it, as well as elements coming in response to point 2 mentioned above. Optionally, more elements as to the different, converging metrics used could be provided to the reader. The general decrease of (each) model performances with elevation could be commented and interpreted, in link with the quality of the interpolations (/extrapolation) of meteorological data and sometimes snow observations at these altitudes.

The setup of the manuscript was discussed with all authors in detail and we found the combination of results and discussion within one chapter as appropriate for this paper. Please also note that referee #1 particularly mentioned the good structure of the paper. We would like to thank for the interest in more interpretations which we could include in this chapter.

Minor Comments

- The last sentence of the abstract overlooks the fact that with altitude, not only the accurate estimation of snowmelt rate gains importance, but also the accurate estimation of SWE, which is one of the hypotheses tested by the paper's set-up.

We will adapt the last sentence of the abstract to: "These findings suggest that with increasing elevation and correspondingly increased contribution of snowmelt to runoff, the accurate estimation of SWE and snowmelt rates gains importance."

References:

Magnusson, J., Gustafsson, D., Hüsler, F., and Jonas, T.: Assimilation of point SWE data into a distributed snow cover model comparing two contrasting methods. Water Resour. Res., 50(10), 7816-7835, doi: 10.1002/2014WR015302, 2014.

Jörg-Hess, S., Fundel, F., Jonas, T., and Zappa, M.: Homogenisation of a gridded snow water equivalent climatology for Alpine terrain: methodology and applications, The Cryosphere, 8, 471-485, doi:10.5194/tc-8-471-2014, 2014.

---

## Author Comment (AC3) · 2 May 2016

Dear Ole Rössler,

we appreciate the positive evaluation of our manuscript and the helpful comments.

Please find below our replies as inserted blue text.

Kind regards,

Nena Griessinger, Jan Seibert, Jan Magnusson and Tobias Jonas

The article addresses a highly relevant topic that is the added value of implementing external snow models and snow data in a hydrological model. In the present study, three different snow models of increasing complexity are attached to a HBV model and tested in 20 different mesoscale catchment within Switzerland. The catchment cover all altitudes present in the Alps. For all catchments the model performance of reproducing the runoff with in the snow melt seasons was assessed and served as the basis to judge the added value of the snow models. The authors found that the implementation of a snow model that additionally assimilates observed SWE data improves the runoff considerably, especially in high altitudes and in snow-rich years. The article is very well structured and written, concise and comprehensive at the time. The article is to my knowledge of original content and suits well in the scope of the journal. I still need to point out some more general concerns and a couple of minor comments. After a revision of the manuscript that take into account this concerns and comments, I would recommend publication:

superior comments/concern:

a) The first concern addresses the interpretation of the results. What is exactly the added value of the assimilated data set. Is it a more sophisticated and correct snow melt model or is it rather the added indirect information of precipitation amounts fallen in high altitudes where the meteorological station network is not present. My interpretation would be the latter, as the differences between model M1 and M2 (e.g. assimilation) are considerable for the highest altitudes. I would appreciate a discussion on this question.

As M1 and M2 differ in the use of the data assimilation algorithm and not in the snow melt model, the added value is based on the information coming from the point snow observations. We will in the revised manuscript in more detail point out the differences between the experiments M1 and M2 to avoid any misunderstanding, and discuss the causes leading to the differences in the results between the two experiments in more depth.

b) A follow up on this issue. The SLF station data are known to overestimate the SWE amounts. How was this issue addressed in the study and if not what are the consequences for your model as you may have calibrated your model against "differently wrong" data.

Following the feedback of referee #2, we will include a discussion about the representativeness and uncertainty of the used punctual snow depth data. The stations used in this study were chosen carefully avoiding sites that are clearly influenced by strong wind drift of snow, frequent sensor failures or known under- or overestimation of snow.

c) The LOO validation produces by nature highly variable performance values. I find it difficult to estimate differences between the models based on medians of boxplot. I would rather use a significance test. I recommend to show validation boxplots side by side and add notches to them.

Thanks for this suggestion. We will consider adapting this visualization of our results.

d) I found examples on the model performance given in Figure 3 and 4 show some room for improvements. Especially in Figure 3 it seems as the threshold for snowmelt was calibrated incorrectly. Is this threshold predefined by the external snowmodel? And if so, doesn't this mean that the snow model itself needs to be updated and calibrated against discharge? And I wonder what the upper benchmark model would look like.

The threshold (horizontal line) indicates 1.5 times the mean observed runoff during the snowmelt season of each year and is not predefined by the external snow model.

special comments/questions:

Page 1 Line 1: Abstract: The first sentence is somehow isolated from the rest of the text. I recommend to delete this sentence

We will delete this sentence.

P2 L1: and the erroneous precipitation input data at higher altitudes?

Thanks, we will add this and include references accordingly.

P3 L 32 "rain input" : which precipitation data set drives the snow model? Also the RHiresD?

Yes, daily RhiresD data were used as input to our snow model. We will clarify this.

P4 L1 ff: Is it correct that all model combinations HBV+M1-M3 as well as upper and lower benchmark models are calibrated? This is somehow suggested by Figure 5. In the calibration section I understood that a calibration was done for M3, upper and lower benchmark.

Yes, all combinations were calibrated. We will clarify this.

P5 L 2-3 what do you mean by "optimal interpolation approach". What magnitude of summed corrections can be found?

Thanks for your question. We refer to the mentioned study of Magnusson et al. (2014).

P5 L12: . . ... , but the RHiresD precipitation data set. Correct?

Yes, also here RhiresD was used as input to the snow model.

P7 L12ff and Figure4: However, the differences between M1-M3 are rather small for the snowmelt season as also indicated by the differences in NSE

Yes, we agree that the differences are rather small.

P7 L27: I wonder if the differences of the LOO validation are significant given the relatively large spread. (see general comments)

Thanks for your recommendation. We will discuss this in the revised version.

P7 L31 and Figure 5: - The benchmark lines are only the median of their respective boxplots? What is the spread of benchmark models? - The only difference between the benchmark model and M3 is a predefined DDF in M3 (cp.P5, L17-18)? Or are there further differences? If not, it is unexpected to see M3 to reach higher performance values then the upper benchmark. - Why is the performance of the benchmark model so weak in comparison to the other models especially in the lowest catchment class where snow does not really play a role?

Thank you for this comment that was also raised by referee #2.

There are further differences that lead to the performances seen here:

Dealing with liquid water content, refreezing, cold content dynamics and the partitioning of rain and snow are - among others - elements that influence the performance of temperature-index models. These elements differ between [M1, M2, M3] and HBV. In [M1, M2, M3] the representation of those processes have been particularly trained for optimal performance in the Swiss Alps.

Further, calibrating HBV for the melt season only – as done in our study – could result in a DDF that is too high during the snow accumulation period. The consequence might be an unbalanced performance with good snowmelt rates during the melt season at the price of too little accumulation earlier in the year with unwanted side effects on the snow depletion dynamics. M3 features a more moderate DDF of 2.5 mm°C$^{-1}$day$^{-1}$ allowing for a more balanced performance over the entire snow season.

We will adapt the manuscript and include this discussion in the revised version.

P8 L17: Please specify snow-rich: extreme snow years do not necessary result in an increased flood risks. To my understanding, largest snow melt contribution to runoff is expected if snow-covered area is largest and snow depth is widely insignificant (if SWE is above a certain minimum).

Thank you, we will clarify and discuss this issue in the revised manuscript.

P8 L30: in snow rich years the extent of snow in the lowlands is presumable larger then in snow-poor years. Accordingly, I also expected an effect of snow-rich years in the lowlands? Can you comment on this?

We did not analyze the performances for single years in the lowlands. Thanks for this remark which we will consider in the revised version.

P9 Conclusion: see superior comments

Figure 1: The blue lines on black are nearly invisible. Please change colors.

Thanks, will be changed.

Figure 2: Instead of showing one specific year, I would rather see a mean snow melt sum. In addition, maps showing differences between the models would increase readability.

Thanks for your recommendation. We discussed showing either cumulative sums or differences between the models with all authors in detail and we found this visualization appropriate for this paper.

Figure 3: Please indicate which model version is represented by the red dashed line.

Figure 3 serves only as graphical explanation of how to calculate $E_{PF}$ and therefore the model version used here is not of importance.

Figure 4: Please add upper benchmark model

We will adapt the figure 4.

Table 1: Instead of numbers I would prefer to see the names of the catchments

We agree and will change the table.

References:

Magnusson, J., Gustafsson, D., Hüsler, F., and Jonas, T.: Assimilation of point SWE data into a distributed snow cover model comparing two contrasting methods. Water Resour. Res., 50(10), 7816-7835, doi: 10.1002/2014WR015302, 2014.

---

## Author Response (AR1)

Dear Dr. Ross Woods,

We would like to thank you and each of the reviewers for their helpful comments that have helped to improve this work.

Please find our discussion of the revisions we have made in response to the comments from the three reviewers in the following pages. You will find specific replies to comments along with references to changes made in the manuscript. The manuscript with all changes marked in red is below this reply letter. The line numbers given in the specific replies refer to the manuscript included below.

For clarification, the manuscript separately uploaded has slightly different line numbers due to the LaTeX journal style template.

Thank you and with best regards,
Nena Griessinger (on behalf of the authors)

**Comment by reviewer:** The study evaluates the value of external snow distributed input into a conceptual hydrologic model in alpine basins. Three different settings are compared for 20 basins in Switzerland. The results show that assimilation of snow improves the runoff model efficiency in basins with mean basin elevation above 2000 m a.s.l.

This is a nice compact study, I enjoyed reading it. The manuscript is clearly written, has a good structure and it is within the scope of the journal.

I have only a few minor comments which might be considered for revision. These include:

1) Introduction: I believe, there are some more relevant studies looking on the benefits of additional snow data in hydrologic model calibration or modelling. Please consider to extend the introduction section accordingly. (please see e.g. Udnaes et al.,2007, Parajka et al. 2007, 2008, or review in Parajka and Blöschl, 2012)

**Answer by authors:** Thank you for the suggested literature; we have included one of the references in the introduction.

**Changes:** Additional literature included in the introduction (see lines 62 to 64).

**Comment by reviewer:** 2) Objectives: Is it the sensitivity (or runoff model efficiency) of conceptual hydrologic model to snow inputs, which is the main objective?

**Answer by authors:** We have clarified our objectives and added text in both, the abstract and the introduction.

**Changes:** Clarified at the end of the introduction (line 67) and in the abstract (line 12).

**Comment by reviewer:** 3) P.3, l.9-10: "daily average values for the entire study"? Please clarify.

**Answer by authors:** For each catchment, we used hourly data aggregated to daily sums to match the temporal resolution of the involved models.

**Changes:** Clarified in Section 2 (see lines 81 to 82).

**Comment by reviewer:** 4) Results: It seems that the way DDF is estimated does affect the performance. Please consider to provide/discuss more detailed information about the sinusoidal function and snow density model used. Does it change with the elevation of the basins?

**Answer by authors:** We added more information on the DDF in the model description (see lines 152 to 154) and extended the discussion by analyzing the performance of the models with particular regards to differences in the formulation of the DDF (see lines 263 to 269).

**Changes:** Additional information as mentioned in the answer above.
* * *
**Comment by reviewer:** 5) Model efficiency: I would suggest to consider extending results and showing also runoff model efficiency (NSE) for the entire calibration/validation periods (not only the selected snowmelt seasons). This might serve as a baseline for comparison with other studies, as well as to allow to evaluate the value of improved snowmelt for the following seasons (e.g. are the soil moisture states/and hence runoff generation different for the three variants?).

**Answer by authors:** The main objective of this study is to assess how different methods for simulating the snow cover influences runoff predictions. Therefore, we focus on a period spanning 60 days that is strongly affected by snowmelt. We agree that improved modeling of snowmelt might also affect runoff model performances later in the year. In response to your comment we performed an additional analysis of the leave-one-out experiment for a snowmelt period of 120 instead of 60 days (see below Figure 1 and Figure 2). This analysis gave similar results with the same relative differences between M1, M2, M3, but with a lower overall performance due to the decreasing relevance of snowmelt as the snow-covered area declines. We thank you for your suggestion, but given these results, we decided not to add the additional figures, but rather include these findings in text-form only.

**Changes:** Extended discussion in Section 4.4 (lines 316-318).

[Figure]

**Figure 1: Results of the leave-one-out approach calculated for 120 days of melt. $E_{PF}$ (left panel) and $E_Q$ (right panel) for each elevation class and snowmelt model. For the individual elevation classes and melt models, the left box plots (darker colors) show the results for the calibration period, and the right box plots (lighter colors) show the results for the validation period. The whisker boxes represent the median (center line), the interquartile range (25-75th percentile; box outline) and highest/lowest performance within the interquartile range +/- 1.5 times of the interquartile range (whiskers). The benchmark performance is denoted by a solid red line (upper benchmark) and a dashed red line (lower benchmark), and the latter only displayed if within the range of the axis limits.**

[Figure]

**Figure 2: Results of the leave-one-out approach calculated for 120 days of melt for catchments with mean elevation above 2000 m.a.s.l. Median (solid lines) and interquartile (25-75th percentile, shading) range of EPF (left panel) and EQ (right panel) for validation years ordered from snow-poor (index=1) to snow-rich (index=15) years.**
* * *
**Comment by reviewer:** 6) Figure 2: Please consider to decrease the legend and increase the size of the maps.

**Answer by authors:** We revised the mentioned figure.

**Changes:** Adapted Figure 2.
* * *
**Comment by reviewer:** 7) Table 1. Please give names to basins.

**Answer by authors:** We revised the mentioned table.

**Changes:** Added catchment names in Table 1.
* * *
**Comment by reviewer:** First, the uncertainty associated with snow depth observation data is never mentioned. As I understand from the manuscript the collected snow depth data were rather punctual and to me, the mentioned 'flatness' of the terrain where they were collected does not guaranty their 'local' representativity. Elaboration on that, and precisions as to the snow depth measurement protocol, would be welcome. An ancillary aspect also regards the hydrological data, which are subject to quite high uncertainties in mountain catchments as a result of frequent shifts in the topography of the river beds. This aspect should at least be discussed.

**Answer by authors:** Thank you for raising the question about the representativeness of the snow observations, which is very relevant. Indeed, flat field observations do not necessarily represent areal mean values over complex terrain as snow accumulation rates are generally smaller in steep terrain as compared to rates over flat terrain. In fact, our snow models do account for the influence of topography on snow distribution and redistribution in mountainous terrain, which is now mentioned in the model description (lines 137 to 147). Furthermore, please note that we have carefully selected our snow data to avoid assimilating data from sites that were influenced by wind or frequent sensor failures, or known to systematically deviate from representative measurements (lines 87 to 92). Regarding the hydrological data, we rely on the plausibility check done by FOEN (Federal Office of Environment), to which we refer (see lines 79 to 82).

**Changes:** Additional information as mentioned in the answer above.
* * *
**Comment by reviewer:** Second, in most calibration and validation sets of simulations, M3 outperforms the upper-benchmark, which relies on a calibrated degree-day factor whereas M3 relies on a constant degree-day factor for all catchments. To me this result is quite counter-intuitive and deserves an explanation.

**Answer by authors:** Thank you for your remark, we agree that this finding may appear counterintuitive and should be discussed. However, please note that all snow models (incl. M3) have been particularly trained for an optimal performance in the Swiss Alps, i.e. regarding the representation of processes like liquid water content, refreezing, cold content dynamics, the partitioning of rain and snow, and redistribution of snow in steep terrain. Furthermore, calibrating HBV for the melt season only could result in a DDF that is too high during the snow accumulation period, which would inhibit an accurate timing of the meltwater release (c.f. updated Figure 4).
We have adapted the manuscript accordingly and provided this discussion to the reader.

**Changes:** Adapted in Section 4.2 (see lines 263 to 269).

**Comment by reviewer:** Finally, a distinct 'discussion' part could be inserted in the manuscript : Section 4.4 after line 11 could be part of it, as well as elements coming in response to point 2 mentioned above. Optionally, more elements as to the different, converging metrics used could be provided to the reader. The general decrease of (each) model performances with elevation could be commented and interpreted, in link with the quality of the interpolations (/extrapolation) of meteorological data and sometimes snow observations at these altitudes.

**Answer by authors:** Thank you for your suggestion, however the setup of the manuscript was discussed with all authors in detail and we found the combination of results and discussion within one chapter appropriate for this paper. We would like to point out that Reviewer #1 particularly appreciated the current structure of the paper. The discussion has been extended in response to all other comments.

**Changes:** No changes.
* * *
**Comment by reviewer:** The last sentence of the abstract overlooks the fact that with altitude, not only the accurate estimation of snowmelt rate gains importance, but also the accurate estimation of SWE, which is one of the hypotheses tested by the paper's set-up.

**Answer by authors:** We have adapted the last sentence of the abstract accordingly.

**Changes:** Adapted abstract.

**Comment by reviewer:** a) The first concern addresses the interpretation of the results. What is exactly the added value of the assimilated data set. Is it a more sophisticated and correct snow melt model or is it rather the added indirect information of precipitation amounts fallen in high altitudes where the meteorological station network is not present. My interpretation would be the latter, as the differences between model M1 and M2 (e.g. assimilation) are considerable for the highest altitudes. I would appreciate a discussion on this question.

**Answer by authors:** As M1 and M2 differ in the use of the data assimilation algorithm only, and not in the snow melt model, the added value is based on the information coming from the point snow observations. Consequently, in both the discussion and the conclusions we highlight the value of data assimilation in M1. We have adapted the model description to clarify the exact difference between M1 and M2.

**Changes:** Additional information about M1 and M2 are given in Section 3.2 (see lines 165 to 168).
* * *
**Comment by reviewer:** b) A follow up on this issue. The SLF station data are known to overestimate the SWE amounts. How was this issue addressed in the study and if not what are the consequences for your model as you may have calibrated your model against "differently wrong" data.

**Answer by authors:** Indeed, flat field observations do not necessarily represent areal mean values over complex terrain, as snow accumulation rates are generally smaller in steep terrain as compared to rates over flat terrain. In fact, our snow models do account for the influence topography on snow distribution and redistribution in mountainous terrain, which is now mentioned in the model description (lines 137 to 147). Furthermore, note that we have carefully selected our snow data to avoid assimilating data from sites that were influenced by wind or frequent sensor failures, or known to systematically deviate from representative measurements (lines 87 to 92).

**Changes:** Additional information as mentioned in the answer above.
* * *
**Comment by reviewer:** c) The LOO validation produces by nature highly variable performance values. I find it difficult to estimate differences between the models based on medians of boxplot. I would rather use a significance test. I recommend to show validation boxplots side by side and add notches to them.

**Answer by authors:** Thank you for this suggestion. We have adapted this visualization of our results and added notches to the box plots as suggested.

**Changes:** Updated Figures 5 and 6.
* * *
**Comment by reviewer:** d) I found examples on the model performance given in Figure 3 and 4 show some room for improvements. Especially in Figure 3 it seems as the threshold for snowmelt was calibrated incorrectly. Is this threshold predefined by the external snowmodel? And if so, doesn't this mean that the snow model itself needs to be updated and calibrated against discharge? And I wonder what the upper benchmark model would look like.

**Answer by authors:** Thank you for catching this mistake. Indeed, in this conceptual figure, the horizontal lines erroneously showed the mean runoff instead of 1.5 times the mean runoff. While calculations of EPF were correct, we replaced Figure 3 with a correct version.

**Changes:** Updated Figure 3.
* * *
**Comment by reviewer:** Page 1 Line 1: Abstract: The first sentence is somehow isolated from the rest of the text. I recommend to delete this sentence

**Answer by authors:** We deleted this sentence.

**Changes:** Adapted abstract.
* * *
**Comment by reviewer:** P2 L1: and the erroneous precipitation input data at higher altitudes?

**Answer by authors:** Thank you; we have added this additional information.

**Changes:** Added information plus references in the introduction (see line 36).
* * *
**Comment by reviewer:** P3 L 32 "rain input" : which precipitation data set drives the snow model? Also the RHiresD?

**Answer by authors:** Yes, RhiresD data were used as input to our snow model. Snowmelt and rain input to HBV are subsequently provided by the external snow model versions.

**Changes:** Clarification in Section 3.1 (line 111).
* * *
**Comment by reviewer:** P4 L1 ff: Is it correct that all model combinations HBV+M1-M3 as well as upper and lower benchmark models are calibrated? This is somehow suggested by Figure 5. In the calibration section I understood that a calibration was done for M3, upper and lower benchmark.

**Answer by authors:** Yes, all combinations were calibrated separately. We now explicitly mention this in the text.

**Changes:** Sentence added in Section 3.1 (see lines 127 to 128).
* * *
**Comment by reviewer:** P5 L 2-3 what do you mean by "optimal interpolation approach". What magnitude of summed corrections can be found?

**Answer by authors:** Thank you for your question. Optimal interpolation, sometimes also referred to as statistical interpolation, is a technical term for a data assimilation technique. See Magnusson et al. (2014) for further details, including the magnitude of summed corrections.

**Changes:** No changes.

References:

Magnusson, J., Gustafsson, D., Hüsler, F., and Jonas, T.: Assimilation of point SWE data into a distributed snow cover model comparing two contrasting methods. Water Resour. Res., 50(10), 7816-7835, doi: 10.1002/2014WR015302, 2014.
* * *
**Comment by reviewer:** P5 L12: . . ... , but the RHiresD precipitation data set. Correct?

**Answer by authors:** Yes, also here RhiresD was used as input to the snow model.

**Changes:** No changes.
* * *
**Comment by reviewer:** P7 L12ff and Figure4: However, the differences between M1-M3 are rather small for the snowmelt season as also indicated by the differences in NSE

**Answer by authors:** Yes, we agree that the differences are small in some instances.

**Changes:** A respective comment was added in Section 4.1 (see line 237).
* * *
**Comment by reviewer:** P7 L27: I wonder if the differences of the LOO validation are significant given the relatively large spread. (see general comments)

**Answer by authors:** We changed the style of the boxplots for clarification (see answer to your general comment above)

**Changes:** Updated Figures 5 and 6.
* * *
**Comment by reviewer:** P7 L31 and Figure 5: - The benchmark lines are only the median of their respective boxplots? What is the spread of benchmark models? - The only difference between the benchmark model and M3 is a predefined DDF in M3 (cp.P5, L17-18)? Or are there further differences? If not, it is unexpected to see M3 to reach higher performance values then the upper benchmark. - Why is the performance of the benchmark model so weak in comparison to the other models especially in the lowest catchment class where snow does not really play a role?

**Answer by authors:** Thank you for your comment. We agree that finding instances where even M3 outperforms the upper benchmark model may appear counter-intuitive. Note however that all snow models (incl. M3) have been particularly trained for an optimal performance in the Swiss Alps, i.e. regarding the representation of processes like liquid water content, refreezing, cold content dynamics, the partitioning of rain and snow, and redistribution of snow in steep terrain. Further,

calibrating HBV for the melt season only could result in a DDF that is too high during the snow accumulation period, which would inhibit an accurate timing of the meltwater release (c.f. updated Figure 4).

We have adapted the manuscript and included this discussion in the revised version.

**Changes:** Added information in 4.2 (see lines 263 to 269).
* * *
**Comment by reviewer:** P8 L17: Please specify snow-rich: extreme snow years do not necessary result in an increased flood risks. To my understanding, largest snow melt contribution to runoff is expected if snow-covered area is largest and snow depth is widely insignificant (if SWE is above a certain minimum).

**Answer by authors:** Thank you, we have clarified this issue in the revised manuscript.

**Changes:** Sentence added in Section 4.2 (see lines 285 to 286).
* * *
**Comment by reviewer:** P8 L30: in snow rich years the extent of snow in the lowlands is presumable larger then in snow-poor years. Accordingly, I also expected an effect of snow-rich years in the lowlands? Can you comment on this?

**Answer by authors:** We did not analyze the performances for single years in the lowlands. At low elevations however, a large fraction of the snow-covered area can melt out in less than a week, even for snow-rich years. Only as you move to higher catchments, you increase the correspondence between the mid-winter snow mass and the duration of the main melt season. Since our data analyses were performed over an evaluation period of 2 months duration, it is expected to not find a pronounced difference in model performances between snow-rich and snow-poor years for low-elevation catchments, as the time scales at which these differences matter at low elevation is much smaller than the evaluation period.

**Changes:** No changes.
* * *
**Comment by reviewer:** Figure 1: The blue lines on black are nearly invisible. Please change colors.

**Answer by authors:** We changed the colors from blue to white, which we hope improves the presentation.

**Changes:** Changed colors in Figure 1.
* * *
**Comment by reviewer:** Figure 2: Instead of showing one specific year, I would rather see a mean snow melt sum. In addition, maps showing differences between the models would increase readability.

**Answer by authors:** Thank you for your recommendation. We discussed showing either cumulative sums or differences between the models with all authors in detail and we found this visualization appropriate for this paper.

**Changes:** No changes.
* * *
**Comment by reviewer:** Figure 3: Please indicate which model version is represented by the red dashed line.

**Answer by authors:** Figure 3 serves only as graphical explanation of how to calculate EPF, therefore the model version used here is not of importance.

**Changes:** No changes.
* * *
**Comment by reviewer:** Figure 4: Please add upper benchmark model

**Answer by authors:** Thank you for this suggestion, we adapted Figure 4 accordingly.

**Changes:** Added upper benchmark model in Figure 4.
* * *
**Comment by reviewer:** Table 1: Instead of numbers I would prefer to see the names of the catchments

**Answer by authors:** We adapted the mentioned table as requested.

**Changes:** Added catchment names in Table 1.

[revised manuscript text omitted]

Grünewald, T. and Lehning, M.: Are flat-field snow depth measurements representative? A comparison of selected index sites with areal snow depth measurements at the small catchment scale. Hydrol. Process. 29: 1717-1728. http://dx.doi.org/10.1002/hyp.10295, 2015.

385    Helbig, N., van Herwijnen, A., Magnusson, J., and Jonas, T.: Fractional snow-covered area parameterization over complex topography. Hydrol. Earth Syst. Sci., 19, 1339-1351, doi:10.5194/hess-19-1339-2015., 2015.

Hock, R.: Temperature index melt modeling in mountain areas. J. Hydrol., 282(1), 104-115, 2003.

Irannezhad, M., Ronkanen, A.-K., and Kløve, B.: Effects of Climate Variability and Change on Snowpack Hydrological Processes in Finland. Cold Reg. Sci. Technol., 118, 14-29, 2015.

390    Isotta, F. A., Frei, C., Weilguni, V., Perčec Tadić, M., Lassègues, P., Rudolf, B., Pavan, V., Cacciamani, C., Antolini, G., Ratto, S. M., Munari, M., Micheletti, S., Bonati, V., Lussana, C., Ronchi, C., Panettieri, E., Marigo, G. and Vertačnik, G.: The climate of daily precipitation in the Alps: development and analysis of a high-resolution grid dataset from pan-Alpine rain-gauge data. Int. J. Climatol., 34: 1657–1675. doi:10.1002/joc.3794, 2013.

Joerg-Hess, S., Griessinger, N., and Zappa, M.: Probabilistic Forecasts of Snow Water Equivalent and Runoff in
395    Mountainous Areas. J. Hydrometeorol., 16, 2169-2186, 2015.

Jonas, T., Marty, C., and Magnusson, J.: Estimating the snow water equivalent from snow depth measurements in the Swiss Alps. J. Hydrol., 378(1), 161–167, 2009.

Klemeš, V.: Operational testing of hydrological simulation models, Hydrolog. Sci. J., 31 (1), 13–24, doi: 10.1080/ 02626668609491024, 1986.

400    Krause, P., Boyle, D.P., and Bäse, F.: Comparison of different efficiency criteria for hydrological model assessment. Adv. Geosci., 5, 89-97, 2005.

Kumar, M., Marks, D., Dozier, J., Reba, M., and Winstral, A.: Evaluation of distributed hydrologic impacts of temperature-index and energy-based snow models. Adv. Water Resour., 56, 77-89, 2013.

Lang, H. and Braun, L.: On the information content of air temperature in context of snow melt estimation. IAHS Publ., 190,
405    347-354, 1990.

Lindström, G., Johansson, B., Persson, M., Gardelin, M., and Bergström, S.: Development and test of the distributed HBV-96 hydrological model. J. Hydrol., 201, 272–288, 1997.

Magnusson, J., Gustafsson, D., Hüsler, F., and Jonas, T.: Assimilation of point SWE data into a distributed snow cover model comparing two contrasting methods. Water Resour. Res., 50(10), 7816-7835, doi: 10.1002/2014WR015302, 2014.

410    Magnusson, J., Wever, N., Essery, R., Helbig, N., Winstral, A., and Jonas, T.: Evaluating snow models with varying process representations for hydrological applications. Water Resour. Res., 51(4), 2707-2723, 2015.

Martinec, J. and Rango, A.: Indirect evaluation of snow reserves in mountain basins. IAHS Publ., 602, 111-119, 1991.

Martinec, J., Rango, A., and Major, E.: The Snowmelt-Runoff Model (SRM) User's Manual. NASA Reference Publication 1100, Washington D.C., 118, 1983.

415     Nash, J.E. and Sutcliffe, J.V.: River flow forecasting through conceptual models, part 1 - a discussion of principles, J. Hydrol., 10(3), 282-290, 1970.

Ohmura, A.: Physical Basis for the Temperature-Based Melt-Index Method. J. Appl. Meteorol., 40(4), 753-761, 2001.

Parajka, J., Merz, R., and Blöschl, G.: Uncertainty and multiple objective calibration in regional water balance modelling: case study in 320 Austrian catchments. Hydrol. Process., 21(4), 435-446, 2007.

420     Perrin, C., Michel, C., and Andréassian, V.: Improvement of a parsimonious model for streamflow simulation. J. Hydrol., 279(1), 275-289, 2003.

Priestley, C. H. B. and Taylor, R. J.: On the assessment of surface heat flux and evaporation using large-scale parameters. Mon. Weath. Rev., 100(2), 81-92, 1972.

Semmens, K. A. and Ramage, J. M.: Recent changes in spring snowmelt timing in the Yukon River basin detected by 425     passive microwave satellite data. Cryosphere, 7(3), 905-916. doi: 10.5194/tc-7-905-2013, 2013.

Seibert, J.: Multi-criteria calibration of a conceptual runoff model using a generic algorithm. Hydrol. Earth Syst. Sci., 4(2), 215-224, 2000.

Seibert, J.: Reliability of model predictions outside calibration conditions. Nord. Hydrol., 34(5), 477-492, 2003.

Seibert, J. and Vis, M. J. P.: Teaching hydrological modeling with a user-friendly catchment-runoff-model software package. 430     Hydrol. Earth Syst. Sci., 16(9), 3315-3325, 2012.

Thirel, G., Salamon, P., Burek, P., and Kalas, M.: Assimilation of MODIS Snow Cover Area Data in a Distributed Hydrological Model Using the Particle Filter. Remote Sens., 5(11), 5825-5850, 2013.Vehviläinen, B.: Snow cover models in operational watershed forecasting. Publications of Water and Environment Research Institute 11. National Board of Waters and the Environment, Helsinki, Finland, 1992.

435     Viviroli, D. and Weingartner, R.: The hydrological significance of mountains: from regional to global scale. Hydrol. Earth Syst. Sci., 8(6), 1017–1030, doi: 10.5194/hess-8-1017-2004, 2004.

Viviroli, D., Archer, D.R., Buytaert, W., Fowler, H.J., Greenwood, G.B., Hamlet, A.F., Huang, Y., Koboltschnig, G., Litaor, M.I., López-Moreno, J.I., Lorentz, S., Schädler, B., Schreier, H., Schwaiger, K., Vuille, M., and Woods, R.: Climate change and mountain water resources: overview and recommendations for research, management and policy. Hydrol. Earth Syst. 440     Sci., 15(2), 471–504. doi: 10.5194/hess-15-471-2011, 2011.

Walter, M.T., Brooks, E.S., McCool, D.K., King, L.G., Molnau, M., and Boll, J.: Process-based snowmelt modeling: does it require more input data than temperature-index modeling?. J. Hydrol., 300(1), 65-75, 2005.

Whitfield, P.H.: Is 'Centre of Volume' a robust indicator of changes in snowmelt timing?. Hydrol. Process., 27(18), 2691-2698, 2013.

445     Wiesinger, T., 1993. Accurate measurement of snowfall - development of two innovative precipitation gages based on the analysis of existing errors. Dissertation, Universität Wien, Inst. f. Meteorologie und Geophysik, 1993: 229 pp.

**Table 1. Characteristics of 20 Swiss catchments in this study.**

| Number | Station name | Area [km$^2$] | Min elevation [m.a.s.l.] | Max elevation [m.a.s.l.] | Mean elevation [m.a.s.l.] | Elevation class | Begin snowmelt [month-day] | End snowmelt [month-day] |
|---|---|---|---|---|---|---|---|---|
| EZG 2202 | Ergolz - Liestal | 276 | 305 | 1087 | 577 | 1 | 01-01 | 03-01 |
| EZG 2126 | Murg - Wängi | 77 | 501 | 911 | 640 | 1 | 01-14 | 03-14 |
| EZG 2034 | Broye - Payerne, Caserne d'aviation | 416 | 450 | 1402 | 721 | 1 | 01-14 | 03-14 |
| EZG 2343 | Langeten - Huttwil, Häberenbad | 61 | 592 | 1032 | 757 | 1 | 01-14 | 03-14 |
| EZG 2374 | Necker - Mogelsberg, Aachsäge | 89 | 649 | 1359 | 948 | 1 | 02-14 | 04-14 |
| EZG 2321 | Cassarate - Pregassona | 74 | 286 | 1809 | 954 | 1 | 02-14 | 04-14 |
| EZG 2603 | Ilfis - Langnau | 188 | 699 | 1695 | 1040 | 2 | 02-21 | 04-21 |
| EZG 2634 | Kleine Emme - Emmen | 473 | 440 | 2261 | 1044 | 2 | 02-21 | 04-21 |
| EZG 2179 | Sense - Thörishaus, Sensematt | 355 | 609 | 2028 | 1072 | 2 | 03-01 | 05-01 |
| EZG 2609 | Alp - Einsiedeln | 82 | 845 | 1577 | 1096 | 2 | 02-21 | 04-21 |
| EZG 2409 | Emme - Eggiwil, Heidbüel | 127 | 770 | 2007 | 1296 | 2 | 02-21 | 04-21 |
| EZG 2300 | Minster - Euthal, Rüti | 59 | 918 | 1994 | 1345 | 2 | 03-07 | 05-07 |
| EZG 2203 | Grande Eau - Aigle | 130 | 579 | 2830 | 1546 | 2 | 03-14 | 05-14 |
| EZG 2605 | Verzasca - Lavertezzo, Campiòi | 188 | 546 | 2590 | 1656 | 2 | 03-14 | 05-14 |
| EZG 2276 | Grosstalbach - Isenthal | 43 | 931 | 2682 | 1794 | 2 | 03-14 | 05-14 |
| EZG 2232 | Allenbach - Adelboden | 31 | 1360 | 2587 | 1907 | 2 | 03-14 | 05-14 |
| EZG 2366 | Poschiavino - La Rösa | 17 | 1920 | 3005 | 2316 | 3 | 04-14 | 06-14 |
| EZG 2304 | Ova dal Fuorn - Zernez, Punt la Drossa | 56 | 1797 | 2903 | 2337 | 3 | 04-14 | 06-14 |
| EZG 2327 | Dischmabach - Davos, Kriegsmatte | 42 | 1772 | 2869 | 2349 | 3 | 04-14 | 06-14 |
| EZG 2256 | Rosegbach - Pontresina | 67 | 1833 | 3721 | 2686 | 3 | 05-01 | 07-01 |

[Figure]

450 **Figure 1: Locations of snow observation stations (red stars) and 20 studied catchments (white border lines) in Switzerland.**

[Figure]

Spring 2007: Sum of Snowmelt [mm]

**Figure 2: Cumulative snowmelt during the snowmelt season 2007 as calculated by the snow model method M1 (full model with data assimilation, left), M2 (full model without data assimilation, middle), and M3 (simplified model, right). The sums between the three model methods differ depending on the use of observational snow data assimilation and the use of different DDFs.**

[Figure]

**Figure 3: Graphical explanation of how to calculate $E_{PF}$. The yellow background shows a catchment-specific snowmelt season window within which the efficiency criteria were computed. The horizontal line indicates the threshold of 1.5 times the mean observed runoff (blue line) above which measured peak flow events (blue circles) are detected. Red stars present corresponding events of the simulated runoff (dashed red line). See Sect. 3.3 for details.**

[Figure]

**Figure 4: Observed and modeled runoff for the Dischma catchment for year 1999, as well as water input from snowmelt and rain modeled with method M1. The upper benchmark model BM in red.**

[Figure]

**Figure 5: Results of the leave-one-out approach.** $E_{PF}$ **(left panel) and** $E_Q$ **(right panel) for each elevation class and snowmelt model. For the individual elevation classes and melt models, the left box plots (darker colors) show the results for the calibration period, and the right box plots (lighter colors) show the results for the validation period. The whisker boxes represent the median (center line), the interquartile range (25-75th percentile; box outline) and highest/lowest performance within the interquartile range +/- 1.5 times of the interquartile range (whiskers). The benchmark performance is denoted by a solid red line (upper benchmark) and a dashed red line (lower benchmark), and the latter only displayed if within the range of the axis limits.**

470

[Figure]

475 **Figure 6: Results of the differential split-sample approach. EPF (left panel) and EQ (right panel) for each elevation class and snowmelt model. For the individual elevation classes and melt models, the left box plots (darker colors) show the results for the calibration period, and the right box plots (lighter colors) show the results for the validation period. The whisker boxes represent the median (center line), the interquartile range (25-75th percentile; box outline) and highest/lowest performance within the interquartile range +/- 1.5 times of the interquartile range (whiskers). The**
480 **benchmark performance is denoted by a solid red line (upper benchmark) and a dashed red line (lower benchmark), and the latter only displayed if within the range of the axis limits.**

[Figure]

**Figure 7: Results of the leave-one-out approach for catchments with mean elevation above 2000 m.a.s.l. Median (solid lines) and interquartile (25-75th percentile, shading) range of $E_{PF}$ (left panel) and $E_Q$ (right panel) for validation years ordered from snow-poor (index=1) to snow-rich (index=15) years.**

485

---

## Author Response (AR2)

Dear Dr. Ross Woods,

We would like to thank you and the reviewer for your justified question. Please find our discussion in response to the comment from the reviewer in the following pages. The manuscript is below this reply letter.

For clarification, the manuscript separately uploaded has slightly different line numbers due to the LaTeX journal style template.

Thank you and with best regards,
Nena Griessinger (on behalf of the authors)

**Comment by reviewer:** As you emphasized, the effect of snow depth (e.g. SWE) assimilation is twofold: First, the snowfall amounts are corrected (lines 156-157). Second, snow melt rates and model state variables were corrected (lines 156-157). I would like a differentiated statement on why the M1 model is so much better than the M2 model version. Is it mainly due to the corrected input data, or due to corrected snow melt rates (cp. strong difference between M1 and M2 in accumulated snowmelt figure 2)

To elaborate on this comment:

The applied precipitation input data set RHiresD is suspected to (strongly) misrepresent the precipitation in higher elevations (>2000 m). This is because no or very sparse meteorological station data is available there. If I am correct, M2, in contrast to M1, only rely on RHiresD data, suggesting a strong deviation of snowfall amounts in higher elevations.

You highlighted that M1 and M2 model performance differ considerably, especially in your highest elevation class (>2000 m) – this finding indicates that data assimilation is crucial to overcome limitations in the input data set.

Furthermore, Figure 2 shows strong differences between snowmelt rates M1 and M2. I am not sure if this related also to this highest elevation (it is hard to read in the maps). This figure indicates a crucial need to substantially lower snow melt rates. You replied: ", the added value is based on the information coming from the point snow observations."

• I wanted to know which information of point snow observation is responsible for the added value.

I think this information is very helpful for other studies that want to make use of similar snow melt models or the RHiresD data sets.

I would very much appreciate such a statement and recommend publication after adding this clarification.

**Answer by authors:** As correctly stated, in Switzerland snow fall rates at higher elevations are particularly uncertain. The data assimilation framework of M1 has been set up in a way that allows compensating for errors in the precipitation input data. The same framework, however, can also adjust modelled snowmelt rates if offset from observed melt rates. At catchment scale, both mechanisms are important since snow melt volumes depend on the contributing area as well as the melt rate. Which effect of the data assimilation may be more relevant is case specific and depends, amongst other factors, on the exact input data situation, the snow model including its calibration, the data assimilation framework including its particular setup, the physiography of the catchment, and the meteorological circumstances. This is why attempting to find an answer to this question may not be relevant to the general readership of this journal. However, in response to the reviewer's question, we have tried to quantify the changes as an effect of data assimilation separately for the snow accumulation and melt period. We hope the reviewer might find the below information beneficial for using similar datasets.

Approach: We evaluated all grid cells within our catchments above 2000 m.a.s.l.. For simplicity, we considered a fix accumulation period (Sept-1 to Feb-28) and a fix snow melt period (March-1 to Jun-31). Cumulative differences between M1 and M2 where calculated for both periods separately and compared to the equivalent cumulative snow accumulation - snow melt respectively - as calculated by M1. It turned out that for both periods the differences between M1-M2 were on average on the order of 5% of total snow accumulation - snow melt respectively. The results suggest that, for our specific model application, data assimilation was effective during both, the snow accumulation and the snow melt season.

**Changes:** No changes.
* * *

[revised manuscript text omitted]

380     Grünewald, T. and Lehning, M.: Are flat-field snow depth measurements representative? A comparison of selected index sites with areal snow depth measurements at the small catchment scale. Hydrol. Process. 29: 1717-1728. http://dx.doi.org/10.1002/hyp.10295, 2015.

Helbig, N., van Herwijnen, A., Magnusson, J., and Jonas, T.: Fractional snow-covered area parameterization over complex topography. Hydrol. Earth Syst. Sci., 19, 1339-1351, doi:10.5194/hess-19-1339-2015., 2015.

385     Hock, R.: Temperature index melt modeling in mountain areas. J. Hydrol., 282(1), 104-115, 2003.

Irannezhad, M., Ronkanen, A.-K., and Kløve, B.: Effects of Climate Variability and Change on Snowpack Hydrological Processes in Finland. Cold Reg. Sci. Technol., 118, 14-29, 2015.

Isotta, F. A., Frei, C., Weilguni, V., Perčec Tadić, M., Lassègues, P., Rudolf, B., Pavan, V., Cacciamani, C., Antolini, G., Ratto, S. M., Munari, M., Micheletti, S., Bonati, V., Lussana, C., Ronchi, C., Panettieri, E., Marigo, G. and Vertačnik, G.:

390     The climate of daily precipitation in the Alps: development and analysis of a high-resolution grid dataset from pan-Alpine rain-gauge data. Int. J. Climatol., 34: 1657–1675. doi:10.1002/joc.3794, 2013.

Joerg-Hess, S., Griessinger, N., and Zappa, M.: Probabilistic Forecasts of Snow Water Equivalent and Runoff in Mountainous Areas. J. Hydrometeorol., 16, 2169-2186, 2015.

Jonas, T., Marty, C., and Magnusson, J.: Estimating the snow water equivalent from snow depth measurements in the Swiss

395     Alps. J. Hydrol., 378(1), 161–167, 2009.

Klemeš, V.: Operational testing of hydrological simulation models, Hydrolog. Sci. J., 31 (1), 13–24, doi: 10.1080/ 02626668609491024, 1986.

Krause, P., Boyle, D.P., and Bäse, F.: Comparison of different efficiency criteria for hydrological model assessment. Adv. Geosci., 5, 89-97, 2005.

400     Kumar, M., Marks, D., Dozier, J., Reba, M., and Winstral, A.: Evaluation of distributed hydrologic impacts of temperature-index and energy-based snow models. Adv. Water Resour., 56, 77-89, 2013.

Lang, H. and Braun, L.: On the information content of air temperature in context of snow melt estimation. IAHS Publ., 190, 347-354, 1990.

Lindström, G., Johansson, B., Persson, M., Gardelin, M., and Bergström, S.: Development and test of the distributed HBV-

405     96 hydrological model. J. Hydrol., 201, 272–288, 1997.

Magnusson, J., Gustafsson, D., Hüsler, F., and Jonas, T.: Assimilation of point SWE data into a distributed snow cover model comparing two contrasting methods. Water Resour. Res., 50(10), 7816-7835, doi: 10.1002/2014WR015302, 2014.

Magnusson, J., Wever, N., Essery, R., Helbig, N., Winstral, A., and Jonas, T.: Evaluating snow models with varying process representations for hydrological applications. Water Resour. Res., 51(4), 2707-2723, 2015.

410     Martinec, J. and Rango, A.: Indirect evaluation of snow reserves in mountain basins. IAHS Publ., 602, 111-119, 1991.

Martinec, J., Rango, A., and Major, E.: The Snowmelt-Runoff Model (SRM) User's Manual. NASA Reference Publication 1100, Washington D.C., 118, 1983.

Nash, J.E. and Sutcliffe, J.V.: River flow forecasting through conceptual models, part 1 - a discussion of principles, J. Hydrol., 10(3), 282-290, 1970.

415 Ohmura, A.: Physical Basis for the Temperature-Based Melt-Index Method. J. Appl. Meteorol., 40(4), 753-761, 2001.

Parajka, J., Merz, R., and Blöschl, G.: Uncertainty and multiple objective calibration in regional water balance modelling: case study in 320 Austrian catchments. Hydrol. Process., 21(4), 435-446, 2007.

Perrin, C., Michel, C., and Andréassian, V.: Improvement of a parsimonious model for streamflow simulation. J. Hydrol., 279(1), 275-289, 2003.

420 Priestley, C. H. B. and Taylor, R. J.: On the assessment of surface heat flux and evaporation using large-scale parameters. Mon. Weath. Rev., 100(2), 81-92, 1972.

Semmens, K. A. and Ramage, J. M.: Recent changes in spring snowmelt timing in the Yukon River basin detected by passive microwave satellite data. Cryosphere, 7(3), 905-916. doi: 10.5194/tc-7-905-2013, 2013.

Seibert, J.: Multi-criteria calibration of a conceptual runoff model using a generic algorithm. Hydrol. Earth Syst. Sci., 4(2), 215-224, 2000.

Seibert, J.: Reliability of model predictions outside calibration conditions. Nord. Hydrol., 34(5), 477-492, 2003.

Seibert, J. and Vis, M. J. P.: Teaching hydrological modeling with a user-friendly catchment-runoff-model software package. Hydrol. Earth Syst. Sci., 16(9), 3315-3325, 2012.

Thirel, G., Salamon, P., Burek, P., and Kalas, M.: Assimilation of MODIS Snow Cover Area Data in a Distributed
430 Hydrological Model Using the Particle Filter. Remote Sens., 5(11), 5825-5850, 2013.Vehviläinen, B.: Snow cover models in operational watershed forecasting. Publications of Water and Environment Research Institute 11. National Board of Waters and the Environment, Helsinki, Finland, 1992.

Viviroli, D. and Weingartner, R.: The hydrological significance of mountains: from regional to global scale. Hydrol. Earth Syst. Sci., 8(6), 1017–1030, doi: 10.5194/hess-8-1017-2004, 2004.

435 Viviroli, D., Archer, D.R., Buytaert, W., Fowler, H.J., Greenwood, G.B., Hamlet, A.F., Huang, Y., Koboltschnig, G., Litaor, M.I., López-Moreno, J.I., Lorentz, S., Schädler, B., Schreier, H., Schwaiger, K., Vuille, M., and Woods, R.: Climate change and mountain water resources: overview and recommendations for research, management and policy. Hydrol. Earth Syst. Sci., 15(2), 471–504. doi: 10.5194/hess-15-471-2011, 2011.

Walter, M.T., Brooks, E.S., McCool, D.K., King, L.G., Molnau, M., and Boll, J.: Process-based snowmelt modeling: does it
440 require more input data than temperature-index modeling?. J. Hydrol., 300(1), 65-75, 2005.

Whitfield, P.H.: Is 'Centre of Volume' a robust indicator of changes in snowmelt timing?. Hydrol. Process., 27(18), 2691-2698, 2013.

Wiesinger, T., 1993. Accurate measurement of snowfall - development of two innovative precipitation gages based on the analysis of existing errors. Dissertation, Universität Wien, Inst. f. Meteorologie und Geophysik, 1993: 229 pp.

**Table 1. Characteristics of 20 Swiss catchments in this study.**

| Number | Station name | Area [km$^2$] | Min elevation [m.a.s.l.] | Max elevation [m.a.s.l.] | Mean elevation [m.a.s.l.] | Elevation class | Begin snowmelt [month-day] | End snowmelt [month-day] |
|---|---|---|---|---|---|---|---|---|
| EZG 2202 | Ergolz - Liestal | 276 | 305 | 1087 | 577 | 1 | 01-01 | 03-01 |
| EZG 2126 | Murg - Wängi | 77 | 501 | 911 | 640 | 1 | 01-14 | 03-14 |
| EZG 2034 | Broye - Payerne, Caserne d'aviation | 416 | 450 | 1402 | 721 | 1 | 01-14 | 03-14 |
| EZG 2343 | Langeten - Huttwil, Häberenbad | 61 | 592 | 1032 | 757 | 1 | 01-14 | 03-14 |
| EZG 2374 | Necker - Mogelsberg, Aachsäge | 89 | 649 | 1359 | 948 | 1 | 02-14 | 04-14 |
| EZG 2321 | Cassarate - Pregassona | 74 | 286 | 1809 | 954 | 1 | 02-14 | 04-14 |
| EZG 2603 | Ilfis - Langnau | 188 | 699 | 1695 | 1040 | 2 | 02-21 | 04-21 |
| EZG 2634 | Kleine Emme - Emmen | 473 | 440 | 2261 | 1044 | 2 | 02-21 | 04-21 |
| EZG 2179 | Sense - Thörishaus, Sensematt | 355 | 609 | 2028 | 1072 | 2 | 03-01 | 05-01 |
| EZG 2609 | Alp - Einsiedeln | 82 | 845 | 1577 | 1096 | 2 | 02-21 | 04-21 |
| EZG 2409 | Emme - Eggiwil, Heidbüel | 127 | 770 | 2007 | 1296 | 2 | 02-21 | 04-21 |
| EZG 2300 | Minster - Euthal, Rüti | 59 | 918 | 1994 | 1345 | 2 | 03-07 | 05-07 |
| EZG 2203 | Grande Eau - Aigle | 130 | 579 | 2830 | 1546 | 2 | 03-14 | 05-14 |
| EZG 2605 | Verzasca - Lavertezzo, Campiòi | 188 | 546 | 2590 | 1656 | 2 | 03-14 | 05-14 |
| EZG 2276 | Grosstalbach - Isenthal | 43 | 931 | 2682 | 1794 | 2 | 03-14 | 05-14 |
| EZG 2232 | Allenbach - Adelboden | 31 | 1360 | 2587 | 1907 | 2 | 03-14 | 05-14 |
| EZG 2366 | Poschiavino - La Rösa | 17 | 1920 | 3005 | 2316 | 3 | 04-14 | 06-14 |
| EZG 2304 | Ova dal Fuorn - Zernez, Punt la Drossa | 56 | 1797 | 2903 | 2337 | 3 | 04-14 | 06-14 |
| EZG 2327 | Dischmabach - Davos, Kriegsmatte | 42 | 1772 | 2869 | 2349 | 3 | 04-14 | 06-14 |
| EZG 2256 | Rosegbach - Pontresina | 67 | 1833 | 3721 | 2686 | 3 | 05-01 | 07-01 |

[Figure]

**Figure 1: Locations of snow observation stations (red stars) and 20 studied catchments (white border lines) in Switzerland.**

450

[Figure]

**Figure 2: Cumulative snowmelt during the snowmelt season 2007 as calculated by the snow model method M1 (full model with data assimilation, left), M2 (full model without data assimilation, middle), and M3 (simplified model, right). The sums between the three model methods differ depending on the use of observational snow data assimilation and the use of different DDFs.**

[Figure]

**Figure 3: Graphical explanation of how to calculate $E_{PF}$. The yellow background shows a catchment-specific snowmelt season window within which the efficiency criteria were computed. The horizontal line indicates the threshold of 1.5 times the mean observed runoff (blue line) above which measured peak flow events (blue circles) are detected. Red stars present corresponding events of the simulated runoff (dashed red line). See Sect. 3.3 for details.**

[Figure]

**Figure 4: Observed and modeled runoff for the Dischma catchment for year 1999, as well as water input from snowmelt and rain modeled with method M1. The upper benchmark model BM in red.**

[Figure]

Figure 5: Results of the leave-one-out approach. $E_{PF}$ (left panel) and $E_Q$ (right panel) for each elevation class and snowmelt model. For the individual elevation classes and melt models, the left box plots (darker colors) show the results for the calibration period, and the right box plots (lighter colors) show the results for the validation period. The whisker boxes represent the median (center line), the interquartile range (25-75th percentile; box outline) and highest/lowest performance within the interquartile range +/- 1.5 times of the interquartile range (whiskers). The benchmark performance is denoted by a solid red line (upper benchmark) and a dashed red line (lower benchmark), and the latter only displayed if within the range of the axis limits.

[Figure]

**Figure 6: Results of the differential split-sample approach. EPF (left panel) and EQ (right panel) for each elevation class and snowmelt model. For the individual elevation classes and melt models, the left box plots (darker colors) show the results for the calibration period, and the right box plots (lighter colors) show the results for the validation period. The whisker boxes represent the median (center line), the interquartile range (25-75th percentile; box outline) and highest/lowest performance within the interquartile range +/- 1.5 times of the interquartile range (whiskers). The benchmark performance is denoted by a solid red line (upper benchmark) and a dashed red line (lower benchmark), and the latter only displayed if within the range of the axis limits.**

[Figure]

480

**Figure 7: Results of the leave-one-out approach for catchments with mean elevation above 2000 m.a.s.l. Median (solid lines) and interquartile (25-75th percentile, shading) range of $E_{PF}$ (left panel) and $E_Q$ (right panel) for validation years ordered from snow-poor (index=1) to snow-rich (index=15) years.**

485